# Proximity labeling identifies LOTUS domain proteins that promote the formation of perinuclear germ granules in *C. elegans*

Ian F Price[1,2,3], Hannah L Hertz[1,2], Benjamin Pastore[1,2,3], Jillian Wagner[1,4], Wen Tang[1,2]*

[1]Department of Biological Chemistry and Pharmacology, Columbus, United States; [2]Center for RNA Biology, Columbus, United States; [3]Ohio State Biochemistry Program, Columbus, United States; [4]Department of Molecular Genetics, Ohio State University, Columbus, United States

**Abstract** The germ line produces gametes that transmit genetic and epigenetic information to the next generation. Maintenance of germ cells and development of gametes require germ granules—well-conserved membraneless and RNA-rich organelles. The composition of germ granules is elusive owing to their dynamic nature and their exclusive expression in the germ line. Using *Caenorhabditis elegans* germ granule, called P granule, as a model system, we employed a proximity-based labeling method in combination with mass spectrometry to comprehensively define its protein components. This set of experiments identified over 200 proteins, many of which contain intrinsically disordered regions (IDRs). An RNA interference-based screen identified factors that are essential for P granule assembly, notably EGGD-1 and EGGD-2, two putative LOTUS-domain proteins. Loss of *eggd-1* and *eggd-2* results in separation of P granules from the nuclear envelope, germline atrophy, and reduced fertility. We show that IDRs of EGGD-1 are required to anchor EGGD-1 to the nuclear periphery while its LOTUS domains are required to promote the perinuclear localization of P granules. Taken together, our work expands the repertoire of P granule constituents and provides new insights into the role of LOTUS-domain proteins in germ granule organization.

*For correspondence:
tang.542@osu.edu

Competing interest: The authors declare that no competing interests exist.

## Editor's evaluation

The authors use proximately labeling and genetic experiments to identify and functionally characterize new components of *C. elegans* P granules. The conclusions of the paper are well-supported by the data. This work will be of broad interest to developmental biologists, particularly those interested in the formation and function of germ cells.

## Introduction

Germ cells are unique stem cells that give rise to eggs and sperm, and ultimately to an entire organism. Germ cells of diverse species are characterized by the accumulation of electron-dense and membrane-less structures referred to as germ granules (*Lehtiniemi and Kotaja, 2018*; *Trcek and Lehmann, 2019*; *Voronina et al., 2011*). Germ granules are widely distributed throughout the animal kingdom. For example, they are known as 'P granules' in worms (*Strome and Wood, 1982*), 'nuage and polar granules' in fruit flies (*Mahowald, 1968*), 'intermitochondrial cement' in frogs (*Bilinski et al., 2004*), and 'chromatoid bodies' in mice (*Parvinen, 2005*).

Genetic and molecular analyses uncovered common classes of proteins that are enriched in germ granules across phyla (*Anderson and Kedersha, 2009*; *Trcek and Lehmann, 2019*; *Updike and Strome, 2010*; *Voronina et al., 2011*). These include Piwi proteins and their small RNA cofactors—piRNAs that recognize and silence transposable elements *Aravin et al., 2006*; *Brennecke et al., 2007*; *Lau et al., 2006*; *Vagin et al., 2006*; ATP-dependent helicases such as Vasa proteins that are involved in RNA metabolism (*Gustafson and Wessel, 2010*; *Spike et al., 2008*), and LOTUS-domain proteins (Limkain /MARF1, Oskar and Tudor domain-containing proteins 5 and 7) (*Anantharaman et al., 2010*; *Callebaut and Mornon, 2010*; *Kubíková et al., 2020*). Metazoan LOTUS domain proteins promote germ cell development. For example, *Drosophila Oskar* interacts with Vasa through its LOTUS domain and serves as scaffold for germ plasm assembly (*Jeske et al., 2017*; *Lehmann, 2016*). Mouse TDRD5 and TDRD7 are required for spermatogenesis (*Lachke et al., 2011*; *Smith et al., 2004*; *Tanaka et al., 2011*; *Yabuta et al., 2011*). Recent work revealed the identity of three LOTUS-domain proteins in *C. elegans* and has begun to explore their role in germ granule organization and small RNA biogenesis (*Cipriani et al., 2021*; *Marnik et al., 2021*).

*C. elegans* germ granules—P granules—provide an exceptional in vivo model to study granule formation and function (*Seydoux, 2018*; *Strome and Wood, 1982*; *Updike and Strome, 2010*). Biophysical studies combined with high-resolution microscopy describe P granules as phase-transitioned condensates (*Brangwynne et al., 2009*). P granules are present in germ cells throughout the entire *C. elegans* life cycle. Yet strikingly, they appear in different forms. In the adult gonad where germ cells undergo mitosis and differentiation, abundant P granules are associated with the cytoplasmic face of nuclear pores (*Pitt et al., 2000*; *Sheth et al., 2010*; *Updike et al., 2011*). During oogenesis, P granules detach from the nuclear membrane and become diffuse in the cytoplasm. Passed on to progeny, P granules partition to the posterior of one-cell embryos. During following cell divisions, P granules are segregated into the germline blastomeres and begin to coalesce and attach to the nuclear periphery. P granules ultimately become perinuclear in the primordial germ cells which give rise to the adult germ line (*Updike and Strome, 2010*).

More than 40 protein components are enriched in P granules (*Updike and Strome, 2010*). Genetic analyses have revealed an assembly pathway that involves core proteins including DEPS-1, GLH-1, PGL-1, and IFE-1. DEPS-1 is a nematode-specific protein that is required for GLH-1 accumulation (*Spike et al., 2008*). GLH-1 belongs to a conserved Vasa family that contributes to germ line development and fertility (*Gustafson and Wessel, 2010*; *Spike et al., 2008*). In addition to the conserved DEAD-box helicase domains, GLH-1 and some of its paralogs contain phenylalanine-glycine (FG) repeats which are postulated to promote perinuclear localization of P granules (*Chen et al., 2020*; *Marnik et al., 2019*; *Updike et al., 2011*). PGL-1 contains both RNA binding and dimerization domains and serves as a P granule scaffold protein (*Aoki et al., 2016*; *Kawasaki et al., 1998*). Loss of either DEPS-1 or GLH-1 causes dispersal of PGL-1 into the cytoplasm, suggesting DEPS-1 and GLH-1 act upstream of PGL-1 (*Kawasaki et al., 2004*; *Kawasaki et al., 1998*). IFE-1 is a *C. elegans* homolog of eIF4E, an mRNA cap-binding protein (*Keiper et al., 2000*). IFE-1 and PGL-1 interact directly and the association of IFE-1 with P granules depends on PGL-1 (*Amiri et al., 2001*). So far, P granule components are primarily identified by genetic approaches (*Updike and Strome, 2010*). Due to their perinuclear localization and the nature of membrane-less compartments, P granules cannot be easily purified via a conventional fractionation-based approach. Therefore, P granule composition and the molecular rules underlying its assembly and migration remain largely unknown.

Our current study used a proximity-based labeling method in conjunction with mass spectrometry to define the P granule proteome. This uncovered over 200 protein candidates. We show that EGGD-1 and EGGD-2 (embryonic and germline P granule detached), referred to as MIP-1 and MIP-2 (MEG-3 interacting protein), respectively, in a related study (*Cipriani et al., 2021*), play a key role in promoting perinuclear localization of P granules. EGGD-1 associates with the nuclear periphery and its localization depends on its intrinsically disordered regions (IDRs). EGGD-1 recruits the Vasa protein GLH-1 possibly through its LOTUS domains. Loss of *eggd-1* and *eggd-2* causes detachment of P granules from the nuclear periphery, germ line atrophy, and infertility. Taken together, our findings define the germ granule proteome and shed light on the organization principles of germ granules.

## Results

### A proximity labeling system to enrich P granule proteins

To probe the composition of P granules, we employed a biotin ligase-based proximity labeling approach to label P granule proteins. TurboID—an engineered promiscuous biotin ligase derived from bacterial BirA—generates reactive biotin derivatives that label proteins in close proximity to the enzyme (*Branon et al., 2018*). TurboID is active from 20°C to 25°C, a range of temperature suitable for *C. elegans* cultivation (*Branon et al., 2018*). We thus sought to target TurboID to P granules (*Figure 1A*). To this end, we used CRISPR/Cas9 to introduce TurboID sequences to genomic loci of *deps-1*, *glh-1*, *pgl-1*, or *ife-1*, which encode proteins known to reside in P granules (*Figure 1A and B*).

We first assessed if tagged alleles generate functional proteins by examining the fertility of animals expressing individual TurboID-tagged P granule proteins. Loss of *deps-1*, *glh-1*, *pgl-1*, or *ife-1* results in reduced fertility (*Updike and Strome, 2010*). Under normal growth conditions, wild-type strains produced ~209 progeny/animal. We found that strains expressing DEPS-1::TurboID and TurboID::GLH-1 yielded ~88 and ~167 progeny/animal, respectively (*Figure 1B*). The brood size of *TurboID::ife-1* animals exhibited a large variation. A small portion of animals became completely sterile and each animal on average produced ~77 progeny (*Figure 1B*). The strain expressing PGL-1::TurboID was completely sterile, a phenotype that is more severe than *pgl-1* null mutants (*Figure 1B*; *Kawasaki et al., 1998*). This finding suggests that the expression of TurboID could cause toxicity. We used the healthiest strains, expressing DEPS-1::TurboID or TurboID::GLH-1, for the proximity labeling experiments.

To determine if P granules are properly assembled in *deps-1::TurboID* and *TurboID::glh-1 strains*, we first examined the subcellular localization of PGL-1 and Argonaute protein CSR-1, two well-characterized P granule proteins (*Claycomb et al., 2009*; *Kawasaki et al., 1998*). Using genetic crosses, we generated *deps-1::TurboID*, and *TurboID::glh-1* strains expressing PGL-1::TagRFP or GFP:CSR-1. Similar to wild-type animals, PGL-1::TagRFP and GFP::CSR-1 are primarily perinuclear in both TurboID animals (*Figure 1—figure supplement 1A*).

We next examined biotinylation of proteins to assess the activity of TurboID. Two assays were employed: streptavidin blot analysis of whole-animal lysate and immunofluorescence staining of dissected gonads. For the first assay, we lysed adult animals, prepared protein lysates, and visualized biotinylated proteins using streptavidin–horseradish peroxidase blot analysis. In the lane with the untagged control, we detected a few signals which presumably corresponded to biotinylated endogenous proteins (*Figure 1C*; *Watts et al., 2018*). In strains expressing DEPS-1::TurboID and TurboID::GLH-1, more proteins were biotinylated (*Figure 1C*).

For the second assay, we stained dissected gonads with fluorescently labeled streptavidin to examine the subcellular localization of biotinylated proteins. We observed weak cytoplasmic signals in the stained wild-type gonad (*Figure 1D*, upper panel). In the *deps-1::TurboID* and *TurboID::glh-1* gonads, signals of biotinylated proteins were observed in the cytoplasm, but highly enriched in perinuclear structures (*Figure 1D*, middle and bottom panels). These findings indicate that TurboID can be applied to label proteins in *C. elegans* germ line.

### Proteomic analysis of P granules

We next carried out streptavidin affinity pull-down to enrich TurboID-biotinylated proteins. In brief, adult animal lysate was prepared under a denaturing condition. After incubating with streptavidin beads, samples were washed under stringent and denaturing conditions to reduce nonbiotinylated protein contaminants and enrich proteins covalently tagged by TurboID (*Branon et al., 2018*). We found that biotinylated proteins were depleted from the flow-through and efficiently enriched in the pull-down (*Figure 1—figure supplement 1B*).

Biotinylated proteins from untagged control, *TurboID::glh-1* and *deps-1::TurboID* strains were enriched in three independent biological replicates, and identified by mass spectrometry (*Figure 2A and B*). As compared to untagged control, 155 and 127 proteins were significantly enriched by labeling with TurboID::GLH-1 and DEPS-1::TurboID, respectively (fold change ≥8, p<0.05) (*Figure 2A*). *Supplementary file 1* provides a detailed overview of candidate proteins identified in TurboID strains. Of the combined 204 candidates, 38.2% (78/204) were recovered from both *deps-1::TurboID* and *TurboID::glh-1* strains (*Figure 2B* and *Supplementary file 1*).

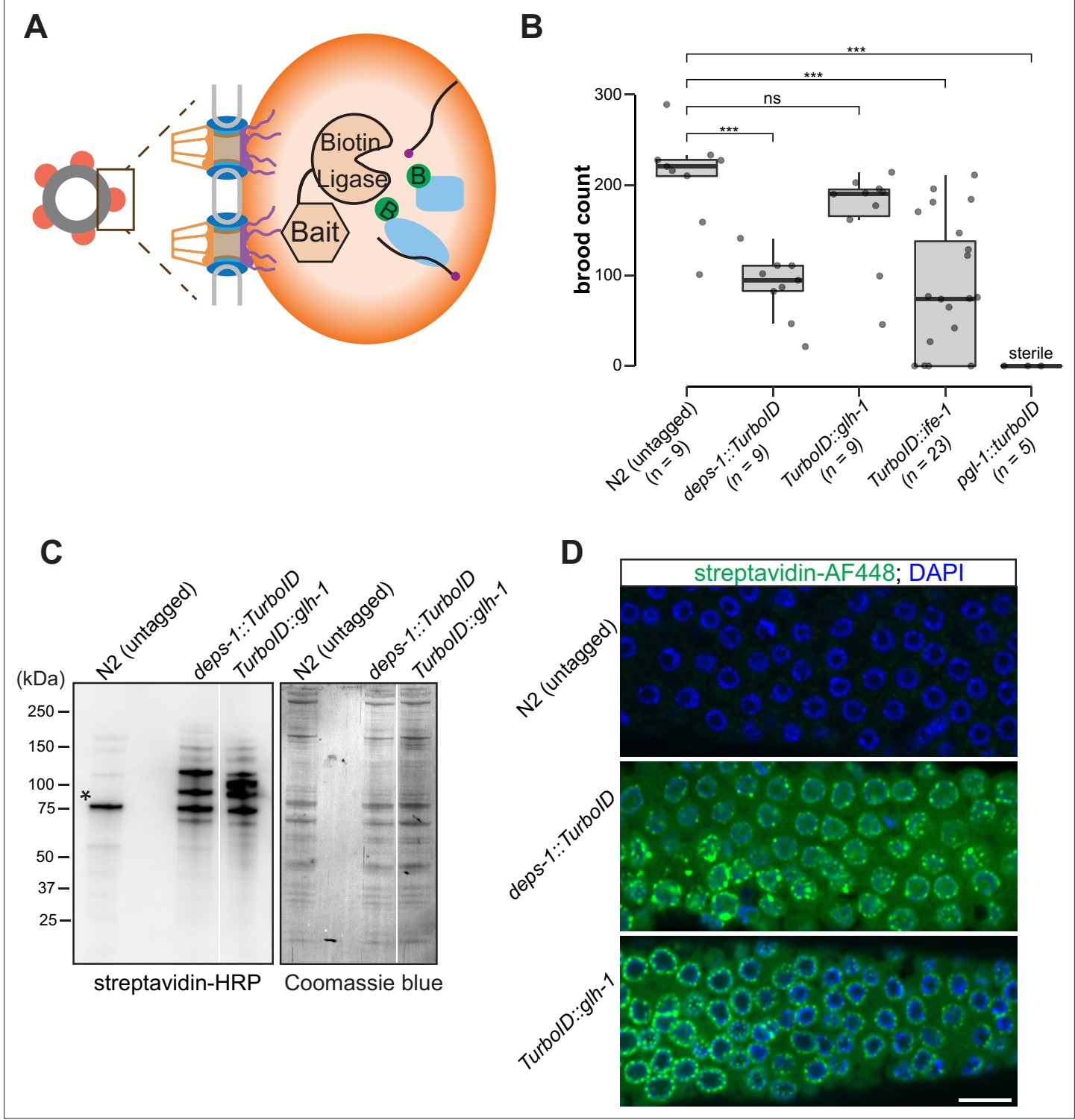

**Figure 1.** A proximity labeling system for specific biotin labeling of P granule proteins. (**A**) Schematic of proximity-based labeling. Known P granule bait proteins are tagged with the promiscuous biotin ligase—TurboID—to label proteins present in P granules. (**B**) Brood sizes of strains endogenously tagged at the loci encoding *deps-1*, *glh-1*, *ife-1*, *pgl-1*, and N2 control. Five independent lines of *pgl-1::TurboID* heterozygotes are sterile. ns: not significant, *** p<0.0005, two-tailed Student's t-test. (**C**) Streptavidin-horseradish peroxidase blotting. The left panel shows whole animal lysates prepared from N2 (untagged control), and strains expressing DEPS-1::TurboID or TurboID::GLH-1 blotted with streptavidin-HRP to visualize biotinylated proteins. The right panel shows Coomassie blue stain of the same membrane. An asterisk marks endogenously biotinylated protein (Based on its size, the protein likely is PCCA-1). (**D**) Streptavidin-Alexa Fluor 488 staining of gonad dissected from N2 (untagged control), and animals expressing DEPS-

*Figure 1 continued on next page*

*Figure 1 continued*

1::TurboID and TurboID::GLH-1. Scale bar=10 µm.

The online version of this article includes the following figure supplement(s) for figure 1:

**Source data 1.** Brood counts of N2(untagged) and TurboID-tagged strains.

**Source data 2.** Uncropped blots of *Figure 1C*.

**Figure supplement 1.** Streptavidin pull-down of biotinylated proteins.

**Figure supplement 1—source data 1.** Uncropped blots of *Figure 1—figure supplement 1B*.

Germ cells in *C. elegans* possess P granules, Z granules, and Mutator foci, three membraneless and perinuclear organelles (*Phillips et al., 2012*; *Strome and Wood, 1982*; *Wan et al., 2018*). Genetic and biochemical approaches identified several components localizing to these three compartments (*Supplementary file 2*; *Manage et al., 2020*; *Updike and Strome, 2010*; *Wan et al., 2018*). Our streptavidin affinity enrichment recovered 90% (18/20) of known P granule proteins and all Z granule proteins—ZNFX-1 and WAGO-4 (*Figure 2B* and *Supplementary file 2*; *Wan et al., 2018*). In contrast, none of the proteins (0/11) in Mutator foci were enriched (*Figure 2B* and *Supplementary file 2*). Our result is consistent with the previous finding that P granules physically contact Z granules, but not Mutator foci (*Wan et al., 2018*).

We inspected candidates identified in both TurboID experiments and found that some are known P granules proteins (*Supplementary file 1*). For example, the PGL family members PGL-1 and PGL-3, and four Vasa family members GLH-1, GLH-2, GLH-3, and GLH-4 were enriched (*Gruidl et al., 1996*; *Kawasaki et al., 2004*; *Kawasaki et al., 1998*; *Spike et al., 2008*). We also identified a subset of Argonaute proteins. These include WAGO-1, WAGO-4, CSR-1, and PRG-1 localized to P granules as well as PPW-1/WAGO-7 and PPW-2/WAGO-3 whose localization requires further investigation (*Batista et al., 2008*; *Claycomb et al., 2009*; *Das et al., 2008*; *Xu et al., 2018*; *Yigit et al., 2006*). Of note, HRDE-1/WAGO-9, a germline nuclear Argonaute protein was not enriched, suggesting TurboID preferentially labeled P granule proteins (*Buckley et al., 2012*). Consistent with the idea that germ granules are hubs for RNA metabolism (*Trcek and Lehmann, 2019*), many factors involved in RNA synthesis, processing, and decay were enriched. These include the Dicer-related helicase DRH-3, Tudor-domain protein EKL-1, and RNA-dependent RNA polymerase EGO-1, terminal nucleotidyl transferase CDE-1, helicase domain containing proteins ZNFX-1 and RDE-12, decapping related proteins DCAP-1 and EDC-4, and 5′–3′ exonuclease XRN-1 (*Ishidate et al., 2018*; *Lall et al., 2005*; *Shirayama et al., 2014*; *Smardon et al., 2000*; *van Wolfswinkel et al., 2009*; *Wan et al., 2018*).

To assess the specificity of proximity labeling, we examined labeled components of nuclear pore complexes. Nuclear pore complexes are among the largest protein structures in cells and comprised of multiple copies of ~30 different proteins known as nucleoporins (Nups) in humans or nuclear pore proteins (NPPs) in *C. elegans* (*Strambio-De-Castillia et al., 2010*; *Updike et al., 2011*). The structure of the nuclear pore complex contains two main functional regions: the central structure which is embedded within the nuclear envelope, and the peripheral structures which extend to both the nuclear interior and cytoplasm termed the nuclear basket and the cytoplasmic filaments, respectively (*Figure 2C*; *Strambio-De-Castillia et al., 2010*). In *C. elegans* germ cells, P granules are associated with the cytoplasmic face of nuclear pore complexes (*Pitt et al., 2000*; *Sheth et al., 2010*; *Updike et al., 2011*). Under stringent purification conditions, we expected to preferentially enrich cytoplasmic-facing NPPs from DEPS-1::TurboID and TurboID::GLH-1 expressing strains. Indeed, components of cytoplasmic filaments NPP-9 and NPP-14, and cytoplasmic ring component NPP-6 were significantly enriched in both TurboID experiments (*Figure 2B and C*, *Supplementary file 3*; *Strambio-De-Castillia et al., 2010*). In summary, we demonstrate that TurboID proximity labeling can be applied to enrich proteins within P granules.

## Properties of the P granule proteome

We next proceeded to characterize the properties of the P granule proteome. First, we conducted gene ontology (GO) enrichment analysis (*Ashburner et al., 2000*; *Raudvere et al., 2019*). As expected, the top three enriched GO terms in the domain of cellular component were 'cytoplasmic ribonucleoprotein granule,' 'ribonucleoprotein granule,' and 'P granule' (*Figure 2D* and *Supplementary file 4*). In the domain of biological process, the top GO terms are 'negative regulation of

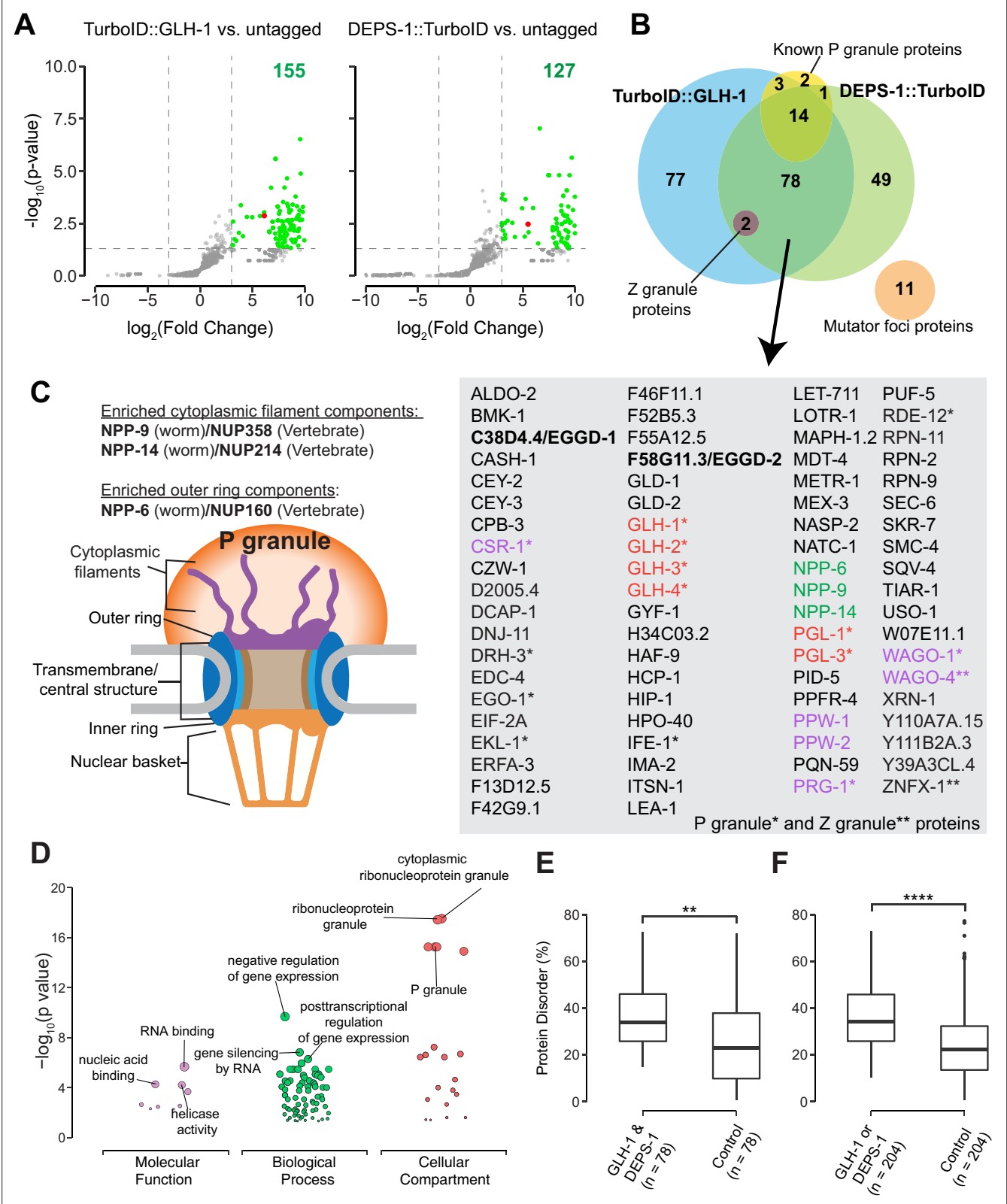

**Figure 2.** Proteomic analysis of P granules and properties of P granule proteome. (**A**) Volcano plots showing statistically significant enriched proteins from strains expressing TurboID::GLH-1 and DEPS-1::TurboID in green. One-tailed Student's t-test, $p<0.05$, $\log_2$(fold change)≥3. GLH-1 and DEPS-1 proteins are shown in red on the respective plots. (**B**) Venn diagram showing overlapping proteins recovered from strains expressing TurboID::GLH-1 and DEPS-1::TurboID, along with previously reported P granule, Z granule, and mutator foci proteins. The list shows proteins enriched by both pull-

*Figure 2 continued on next page*

*Figure 2 continued*

down experiments. Argonaute proteins (purple), nuclear pore proteins (NPPs, green), and core P granule components GLH and PGL-1 family proteins (red). P granule and Z granule proteins are marked with an asterisk and two asterisks, respectively. No reported mutator foci proteins were significantly enriched. (**C**) Schematic of nuclear pores and P granules. Pull-down experiments enrich NPPs (NPP-6, NPP-9, and NPP-14) composing either the cytoplasmic filaments or outer ring (purple). NPPs and their vertebrate homologs are shown. (**D**) Functional gene set enrichment analysis of proteins significantly enriched in both pull-down experiments organized by molecular function, biological processes, and cellular compartments. Top three enriched categories (based on p-value) are labeled. (**E**) Boxplot showing the average disorder of proteins enriched in both *TurboID::glh-1* and *deps-1::TurboID* (n=78) as predicted using IUPRED compared to a random control set (n=78). The average disorder of proteins was derived by comparing the total IUPRED score of each protein to its length. Wilcoxon rank-sum test (p<0.01 \*\*). (**F**) The same analysis as in (**E**), but with proteins labeled in either *TurboID::glh-1* or *deps-1::TurboID* pull-down experiments (n=204). Wilcoxon rank-sum test (p<0.0001 \*\*\*\*).

The online version of this article includes the following figure supplement(s) for figure 2:

**Source data 1.** Normalized spectral counts for N2 (untagged) and TurboID tagged strains.

**Source data 2.** Average IUPred disorder score for each *Caenorhabditis elegans* protein.

**Figure supplement 1.** Network analysis of proteins enriched from TurboID.

gene expression,' 'gene silencing by RNA,' and 'posttranscriptional regulation of gene expression' (***Figure 2D*** and ***Supplementary file 4***). Additionally, GO terms including 'oocyte construction' and 'oocyte anterior/posterior axis specification' were highly enriched (***Supplementary file 4***), consistent with the notion that P granules are essential for gametogenesis and germ line maintenance. In the domain of molecular function, GO terms including 'RNA binding,' 'nucleic acid binding' and 'helicase activity' were significantly enriched, suggesting P granule proteins interact extensively with RNAs (***Figure 2D*** and ***Supplementary file 4***).

We constructed the P granule protein interaction network using publicly available protein-protein interaction (PPI) data (***Jensen et al., 2009***). When examining proteins labeled by both DEPS-1::TurboID and TurboID::GLH-1, we found the resulting network contained 78 nodes and 200 non-redundant edges (***Figure 2—figure supplement 1***). The number of edges was significantly higher than expected by chance (p<1 ×10⁻¹⁶), implying that the proteins are at least partially biologically connected as a group and multiple PPIs may act in P granule assembly. We found that multiple P granule proteins showed high betweenness centrality within the network (***Figure 2—figure supplement 1***). At least three additional clusters were formed (***Figure 2—figure supplement 1***). These clusters consist of NPPs such as NPP-6, NPP-9, and NPP-14 (***Strambio-De-Castillia et al., 2010***); kinetochore components such as HCP-1 (***Cheeseman, 2014***; ***Kitagawa, 2009***); or proteasome components such as RPN-2 (***Marshall and Vierstra, 2019***). Taken together, these findings suggest that known and unknown P granule proteins identified by TurboID form a dense protein interaction network.

Proteins containing IDRs are often found in biomolecular condensates (***Markmiller et al., 2018***; ***Molliex et al., 2015***; ***Nott et al., 2015***). IDRs themselves fail to form stable structures. Yet they participate in multivalent protein–protein, protein–DNA, and/or protein–RNA interactions (***Oldfield and Dunker, 2014***; ***Uversky, 2017***). We next determined if IDR-containing proteins were enriched in the P granule proteome defined by TurboID. We employed the IUPred algorithm which predicts protein disorder by estimating the total pairwise inter-residue interaction energy of amino acids (***Dosztányi et al., 2005***). A probabilistic score of each residue ranging from 0 (complete order) to 1 (complete disorder) was generated. We calculated the sum of probabilistic scores and further normalized it to the protein length (***Supplementary file 1***). Compared to a randomly selected control group the P granule proteome displayed a higher degree of disorder (***Figure 2E and F***). These data suggest that IDR-containing proteins are overrepresented in P granules.

## An RNAi-based screen to identify factors required for P granule formation

P granule proteomic data obtained by TurboID provided a unique opportunity to identify factors that directly participate in P granule assembly. To this end, we have begun to conduct an RNA interference (RNAi)-based screen using a reporter strain expressing PGL-1::TagRFP. Different from a previous genome-wide RNAi screen (***Updike and Strome, 2009***), the reporter in this study had a null allele of *rrf-3*, which renders the strain hypersensitive to RNAi (***Simmer et al., 2002***). We used fluorescence microscopy to search for changes in PGL-1::TagRFP expression in animals in which candidate genes were depleted. Out of 31 genes, we screened so far 11 genes upon depletion caused PGL-1::TagRFP

phenotypes in adult animals (*Figure 3A*). Among these 11 genes, *cpf-2* has been reported to be required for proper PGL-1 localization (*Updike and Strome, 2009*). A recent study showed that depletion of *cey-2* and *cey-3*, two genes encoding Y-box binding proteins, induces PGL-1 aggregation (*Calculli et al., 2021*). We placed PGL-1::TagRFP phenotypes into broad categories including detached from nuclear envelope, reduced expression, diffuse, large aggregates, and none observed (*Figure 3A*). This search identified C38D4.4 which is required for proper PGL-1::TagRFP localization in the pachytene region (*Figure 3B*). Depletion of C38D4.4 resulted in the formation of large PGL-1-containing aggregates, many of which are separated from the nuclear envelope of germ cells. Based on this phenotype, we named C38D4.4 as *eggd-1* for embryonic and germline P granule detached.

F58G11.3 is a predicted paralog of EGGD-1. It is 33.3% identical to EGGD-1 at the amino acid level. Interestingly, F58G11.3 was also identified from TurboID experiments (*Figure 2B* and *Supplementary file 1*). We named F58G11.3 as *eggd-2*, although RNAi against *F58G11.3* did not yield noticeable change in PGL-1::TagRFP localization (*Figure 3A*). Of note, an independent study refers to EGGD-1 and EGGD-2 as MEG-3 interacting proteins MIP-1 and MIP-2, respectively (*Cipriani et al., 2021*).

## EGGD-1 and EGGD-2 promote perinuclear localization of P granules and fertility

Using CRISPR/CAS9 editing, we generated a null allele of *eggd-1* by deleting the full open reading frame. Consistent with the result from RNAi experiments, PGL-1::TagRFP was dispersed into the cytoplasm in *eggd-1* mutants (*Figure 3B and C*). We generated a null allele of *eggd-2* by deleting its full open reading frame, and another allele of *eggd-2* bearing a 17-nucleotide insertion downstream of the start codon. Neither of *eggd-2* alleles yielded noticeable change in PGL-1::TagRFP localization in the pachytene region (*Figure 3B*, *Figure 3—figure supplement 1A*). Thereafter, we further characterized the phenotype of *eggd-2* that bears the 17-nucleotide insertion. In *C. elegans*, germ cell nuclei are situated along the outer surface of the gonadal tube and share a central cytoplasmic core, termed rachis (*Amini et al., 2015*). We inspected PGL-1::TagRFP fluorescence on the surface and core of the germ line. In wild-type animals, fluorescence signal was primarily associated with the periphery of germ cell nuclei. In *eggd-1* mutants, however, fewer PGL-1 foci were perinuclear, and most diffused into the rachis (*Figure 3C*). Loss of *eggd-2* appeared to have a minor effect on PGL-1 localization (*Figure 3C*). Deletion of both *eggd-1* and *eggd-2* caused dispersal of perinuclear PGL-1::TagRFP and accumulation of large cytoplasmic PGL-1 aggregates (*Figure 3C*). We used ImageJ to quantify PGL-1::TagRFP signal at the germline edge and rachis (*Figure 3—figure supplement 1B*). Compared to that in wild-type, the rachis/edge ratio was increased in *eggd-1* mutants, and further increased in *eggd-1; eggd-2* double mutants (*Figure 3D*).

During the first embryonic cell division, P granules are partitioned to the germ lineage of embryos (*Strome and Wood, 1982*). During following cell divisions, they are selectively eliminated in somatic cells, and begin to coalesce and attach to the nuclear periphery of germ cells (*Seydoux, 2018*; *Updike and Strome, 2010*; *Zhang et al., 2009*). Consistent with previous findings (*Strome and Wood, 1982*), PGL-1::TagRFP foci were detected in germ cells as well as in somatic cells around 28 cell stage wild-type embryos. By the comma stage, PGL-1::TagRFP became predominantly perinuclear (*Figure 3—figure supplement 1C*). In contrast, PGL-1 failed to concentrate in the germ lineage in *eggd-1*, *eggd-2*, and *eggd-1; eggd-2* embryos (*Figure 3—figure supplement 1C*). Furthermore, perinuclear P granules were not formed in comma stage embryos upon loss of *eggd-1* and *eggd-2* (*Figure 3—figure supplement 1C*). Taken together, these findings suggest that EGGD-1 and EGGD-2 promote the perinuclear localization of P granules in both adult germ lines and embryos.

Disruption of germ granule formation causes defects in germ line development and infertility in diverse organisms (*Anderson and Kedersha, 2009*; *Trcek and Lehmann, 2019*; *Voronina et al., 2011*). Using PGL-1::TagRFP as a germ cell marker, we examined whether *eggd-1* and *eggd-2* mutants exhibit defects in germ line proliferation. Both *eggd-1* and *eggd-2* mutants displayed a diminutive germ line compared to wild-type animals (*Figure 3E*, *Figure 3—figure supplement 1D*). Additive genetic effects were observed in *eggd-1; eggd-2* double mutants (*Figure 3E*, *Figure 3—figure supplement 1D*).

We next evaluated the fertility of *C. elegans* strains upon loss of *eggd-1* and/or *eggd-2* using two approaches. In the first approach, we outcrossed mutants with wild-type animals and measured the

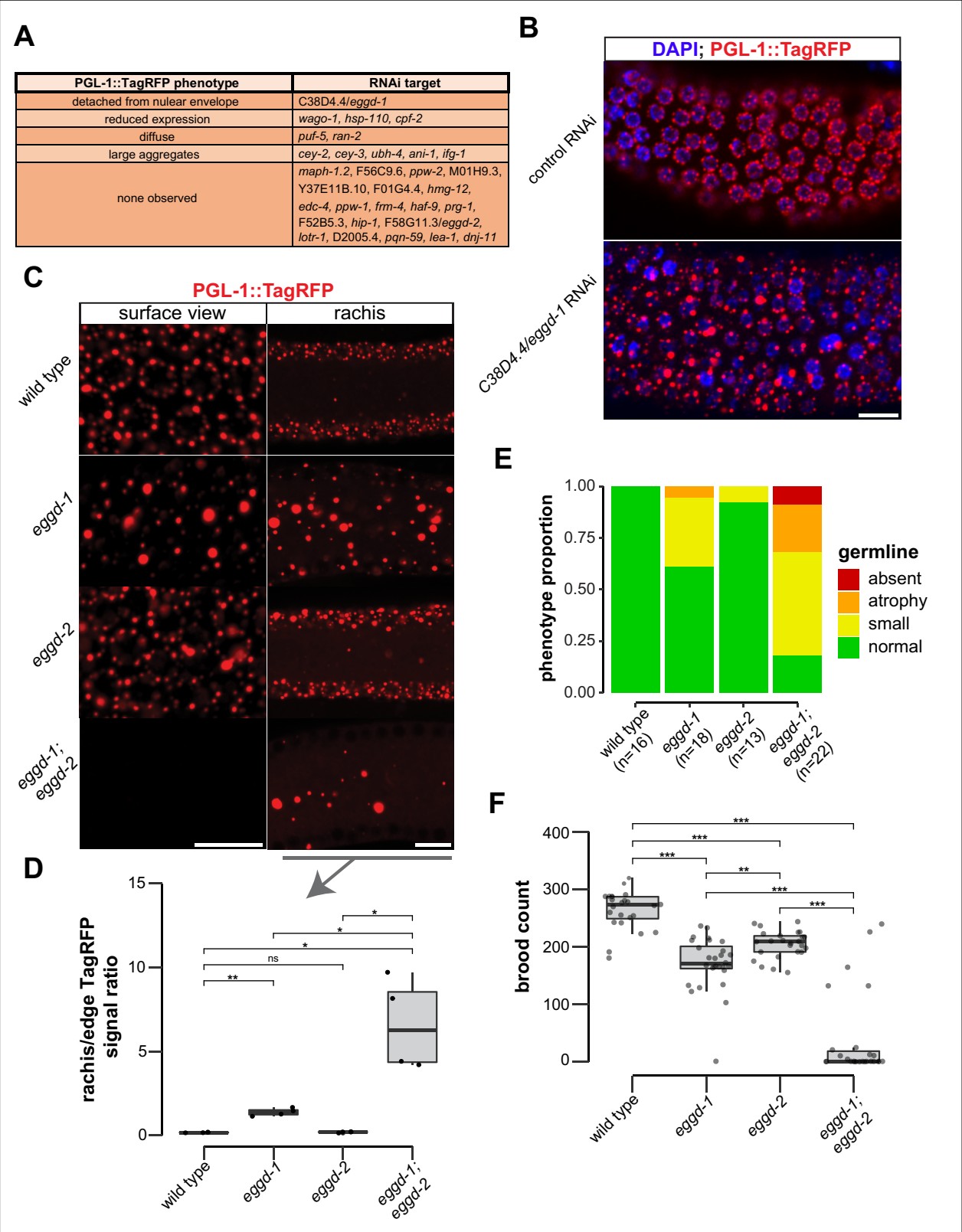

**Figure 3.** EGGD-1 and EGGD-2 promote perinuclear localization of P granules. (**A**) PGL-1::TagRFP phenotypes. Summarized PGL-1::TagRFP phenotypes from an RNAi-based screen. RNAi targets are grouped by the observed PGL-1::TagRFP phenotypes in *rrf-3; pgl-1::TagRFP* adult germ lines. (**B**) Fluorescence micrographs of dissected gonads from *pgl-1::TagRFP* reporter animals after two successive generations of control RNAi or *C38D4.4/eggd-1* RNAi. Scale bar=10 μm. (**C**) Single confocal slices of the edge and rachis of the germ line in live adult animals expressing PGL-

*Figure 3 continued on next page*

Figure 3 continued

1::TagRFP. Wild-type animals, and animals bearing mutations in *eggd-1, eggd-2,* and *eggd-1; eggd-2* are shown. Scale bar=10 µm. Images are representative of at least four animals. (**D**) Boxplot of quantified rachis versus edge PGL-1::TagRFP signal intensity in *eggd-1, eggd-2,* and *eggd-1; eggd-2* mutant backgrounds. ns: not significant, *p<0.05, **p<0.005, two-tailed Student's t-test (n=4). (**E**) Germ line morphology in adult *eggd-1, eggd-2,* and *eggd-1; eggd-2* mutants compared to wild-type animals. All strains express PGL-1::TagRFP. Representative images of absent, atrophy, small, and normal classifications are shown in *Figure 3—figure supplement 1D*. (**F**) Brood counts of wild-type, *eggd-1, eggd-2,* and *eggd-1; eggd-2* animals. All strains express PGL-1::TagRFP. **p<0.005, ***p<0.0005, two-tailed Student's t-test.

The online version of this article includes the following figure supplement(s) for figure 3:

**Source data 1.** Gray value measurements used to quantify rachis/edge signal ratio.

**Source data 2.** Categorization of germ line atrophy in *eggd-1, eggd-2,* and *eggd-1; eggd-2* mutants.

**Source data 3.** Brood counts of *eggd-1, eggd-2,* and *eggd-1; eggd-2* mutants.

**Figure supplement 1.** Embryonic PGL-1::TagRFP localization and images of diminutive germ line.

**Figure supplement 1—source data 1.** Percentage of plates with progeny over successive generations.

brood sizes of outcrossed strains at approximately generation 12. The wild-type strain produced ~263 progeny/animal (*Figure 3F*). As compared to wild-type, *eggd-1* and *eggd-2* animals displayed moderate reduction in brood size, generating ~170 and ~204 progeny/animal, respectively. Strikingly, *eggd-1; eggd-2* double mutants exhibited an additive fertility deficit and produced only ~37 progeny/animal (*Figure 3F*). In the second approach, we outcrossed mutant animals with wild-type, tracked 10 lines of *eggd-1, eggd-2,* and double mutants, and scored whether or not each line generated offspring every two generations. Wild-type animals were fertile in the course of the experiment (~21 generations) (*Figure 3—figure supplement 1E*). Animals deficient for EGGD-1 or EGGD-2 exhibited decline in fertility over generations (*Figure 3—figure supplement 1E*). The *eggd-1; eggd-2* double mutants became sterile more rapidly when compared to single mutants (*Figure 3—figure supplement 1E*). Altogether, these observations suggest that EGGD-1 and EGGD-2 are required for the maintenance and immortality of *C. elegans* germ line.

## EGGD-1 and EGGD-2 contain two IDRs and two putative LOTUS domains

We next characterized EGGD-1 and EGGD-2 amino acid sequences. First, both EGGD-1 and EGGD-2 were relatively disordered (*Supplementary file 1*). Based on IUPred algorithm, two IDRs were identified in EGDD-1 and EGGD-2, one close to their N-termini and the other located at their C-termini (*Figure 4A and B*; *Dosztányi et al., 2005*).

We next searched for conserved domains within EGGD-1 and EGGD-2 using the HHpred program (*Zimmermann et al., 2018*). HHpred is one of the most sensitive methods for remote homology detection (*Zimmermann et al., 2018*). The HHpred search identified two regions in EGGD-1 and EGGD-2 that are homologous to the LOTUS domain of *D. melanogaster* Oskar and *H. sapiens* TDRD5 and TDRD7. LOTUS domains are divided into two subclasses depending on the absence or presence of a C-terminal extension: minimal LOTUS (mLOTUS) and extended LOTUS (eLOTUS) which contains an extra C-terminal alpha-helix (*Jeske et al., 2017*). Similar to the LOTUS domains in Oskar, TDRD5 or TDRD7, LOTUS domains in EGGD-1 and EGGD-2 are predicted to contain the C-terminal alpha-helix extension (*Figure 4—figure supplement 1A*; *Anantharaman et al., 2010*; *Callebaut and Mornon, 2010*; *Kubíková et al., 2020*). We conclude that EGGD-1 and EGGD-2 harbor four domains: two eLOTUS domains and two IDRs (*Figure 4A and B*).

## Roles of EGGD-1 LOTUS domains and IDRs in perinuclear P granule formation

We next examined EGGD-1 and EGGD-2 protein localization. CRISPR/CAS9 was used to generate fluorescently tagged alleles of *eggd-1* and *eggd-2*. Both proteins are expressed in the adult germ line and germ lineage of embryos (*Figure 4—figure supplement 1B and C*). Consistent with the TurboID data, both EGGD-1::GFP and mCherry::EGGD-2 localize to perinuclear P granules (*Figure 4—figure supplement 1B and C*). Of note, the fluorescent signal of EGGD-2 was much weaker than EGGD-1::GFP or PGL-1::TagRFP. Considering EGGD-1 has a stronger impact on P granule assembly in the adult germ line (*Figure 3C*), we decided to focus on EGGD-1 for further characterization.

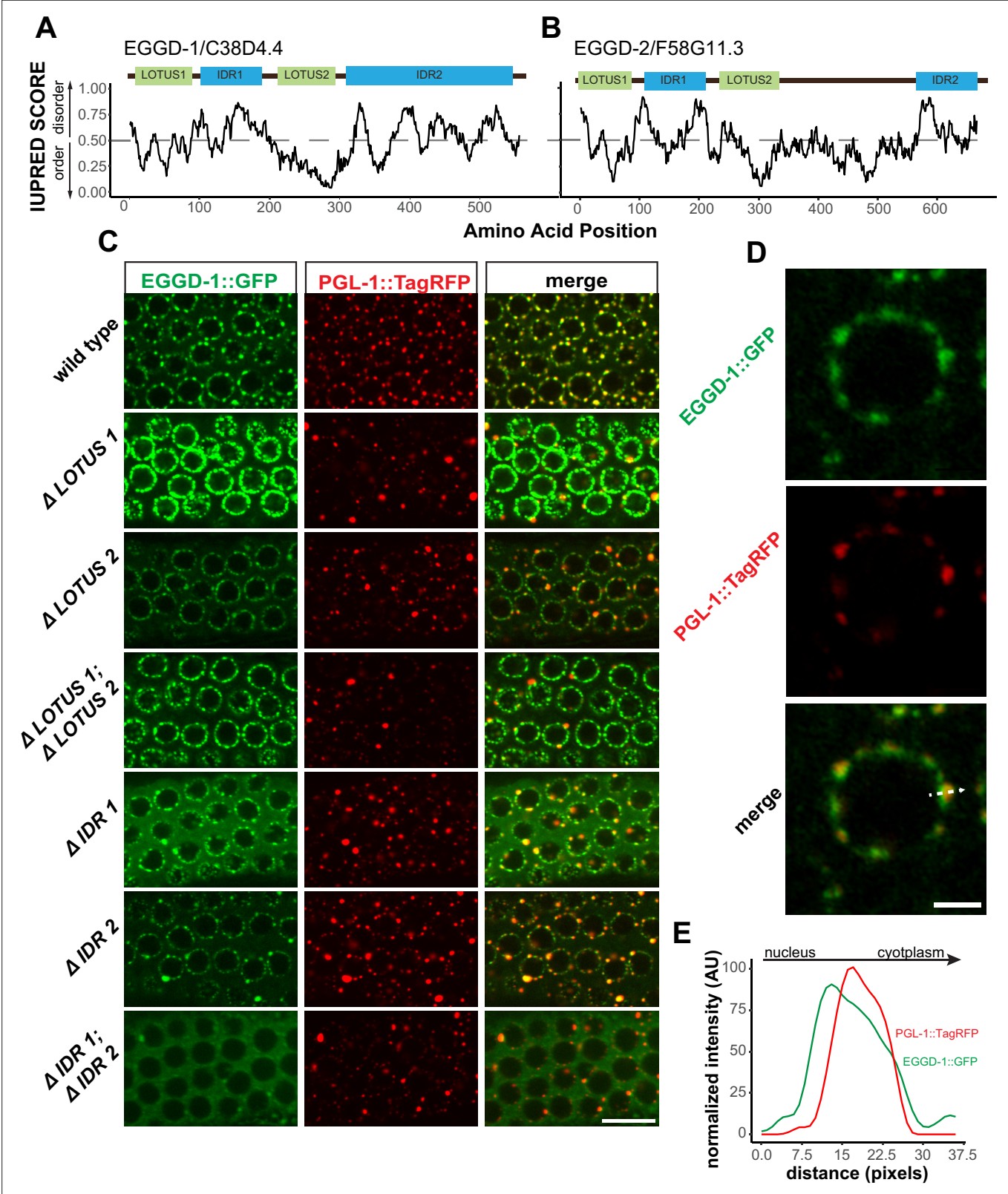

**Figure 4.** EGGD-1 and EGGD-2 contain IDRs and LOTUS domains. (**A, B**). Domain architecture and IUPRED score for EGGD-1 (**A**) and EGGD-2 (**B**). Regions of proteins with an IUPRED score above 0.5 are predicted to be disordered, while regions below 0.5 are predicted to be ordered. Intrinsically disordered regions are shown in blue and abbreviated as 'IDR.' LOTUS domains are shown in green. (**C**) Spinning disc confocal images (100× objective) showing the localization of EGGD-1::GFP protein, and a series of EGGD-1::GFP domain deletion mutants in pachytene germ cells of live animals

*Figure 4 continued on next page*

*Figure 4 continued*

expressing PGL-1::TagRFP. Images are representative of over five animals. Scale bar=10 µm. (**D**) Super-resolution Zeiss Airyscan image of a single pachytene nucleus in animals co-expressing EGGD-1::GFP and PGL-1::TagRFP. Scale bar=2 µm. (**E**) Intensity profile of EGGD-1::GFP and PGL-1::TagRFP signals along the dotted arrow in panel (**D**). AU, arbitrary unit; IDR, intrinsically disordered region.

The online version of this article includes the following figure supplement(s) for figure 4:

**Source data 1.** IUPred score for each amino acid in EGGD-1 and EGGD-2.

**Source data 2.** Gray value intensity along the dotted line in *Figure 4D*.

**Figure supplement 1.** Localization of EGGD-1 and EGGD-2 proteins.

To interrogate the role of IDRs and LOTUS domains, we employed CRISPR/CAS9 to generate a series of *eggd-1* alleles that delete individual IDRs, individual LOTUS domains, both IDRs, or both LOTUS domains (*Figure 4C*). We next examined the localization of EGGD-1 and PGL-1::tagRFP in live animals using spinning disk confocal microscopy. In the adult germ line, full-length EGGD-1::GFP partly co-localized with PGL-1::TagRFP foci (*Figure 4C*). Super-resolution imaging revealed that EGGD-1::GFP foci surround PGL-1::TagRFP foci in some instances, but largely are enriched at the base of P granules (*Figure 4D and E*).

When the first LOTUS domain was deleted (ΔLOTUS 1), EGGD-1 displayed prominent perinuclear localization. In contrast, PGL-1::tagRFP was mislocalized. Upon removal of the second LOTUS domain (ΔLOTUS 2), EGGD-1 localized to the nuclear periphery while perinuclear PGL-1 foci were modestly lost. Deletion of both LOTUS one and LOTUS two did not appear to affect EGGD-1 localization. However, PGL-1 became predominantly cytoplasmic (*Figure 4C*). When the first IDR was deleted (ΔIDR 1), perinuclear EGGD-1 foci remained, but some EGGD-1 became dispersed into the cytoplasm. When the second IDR was deleted (ΔIDR 2), both EGGD-1 and PGL-1::tagRFP foci were largely attached to the nuclear periphery albeit the presence of some large aggregates. When both IDR one and IDR two were deleted, we found that EGGD-1 was almost completely dispersed in the cytoplasm in the pachytene region. And similarly, PGL-1::TagRFP became cytoplasmic (*Figure 4C*). These findings imply that LOTUS domains and IDRs in EGGD-1 have distinct roles in promoting perinuclear P granules: IDRs are required to anchor EGGD-1 protein to the nuclear periphery, and LOTUS domains likely recruit additional P granule proteins.

## EGGD-1 acts upstream of GLH-1 in P granule assembly

Previous studies showed one conserved function of eLOTUS domains is to interact with Vasa (*Jeske et al., 2015*; *Jeske et al., 2017*). *C. elegans* Vasa protein GLH-1 is known to localize to P granules and promote their assembly (*Chen et al., 2020*; *Marnik et al., 2019*; *Updike et al., 2011*). We thus tested the interaction between EGGD-1/2 and GLH-1. In particular, we performed epistasis analyses to order EGGD-1/2 and GLH-1 in the P granule assembly pathway. As expected, GFP::GLH-1 localized to P granules in wild-type animals. However, when *eggd-1* was deleted, GFP::GLH-1 became diffuse and most of the protein was no longer associated with the nuclear periphery (*Figure 5A*). Deletion of *eggd-2* did not appear to affect GFP::GLH-1 localization. There was a striking change in GFP:GLH-1 distribution in *eggd-1; eggd-2* double mutant animals, with the majority of fluorescence signals appearing throughout the cytoplasm (*Figure 5A*). These findings suggest perinuclear GFP::GLH-1 primarily depends on EGGD-1.

We next used RNAi to deplete GLH-1 from EGGD-1::GFP; PGL-1::TagRFP expressing worms. Of note, *glh-1* RNAi likely depleted other GLH family members such as GLH-2 due to the sequence similarity. Consistent with previous studies (*Spike et al., 2008*), PGL-1::TagRFP became diffused into the cytoplasm upon depletion of GLH protein(s). In contrast, EGGD-1::GFP remained at the nuclear periphery (*Figure 5B*). These observations suggest that unlike other P granule proteins, perinuclear localization of EGGD-1 does not rely on the GLH family.

A subset of NPPs, such as NPP-10, localize to P granules and are indispensable for their integrity (*Updike and Strome, 2009*; *Voronina and Seydoux, 2010*). We next sought to determine if EGGD-1 localization depends on NPPs. Indeed, depletion of *npp-10* caused diffuse PGL-1::TagRFP (*Updike and Strome, 2009*; *Voronina and Seydoux, 2010*). Similarly, EGGD-1::GFP became detached from the nuclear membrane and formed aggregates in the cytoplasm upon depletion of *npp-10* (*Figure 5B*).

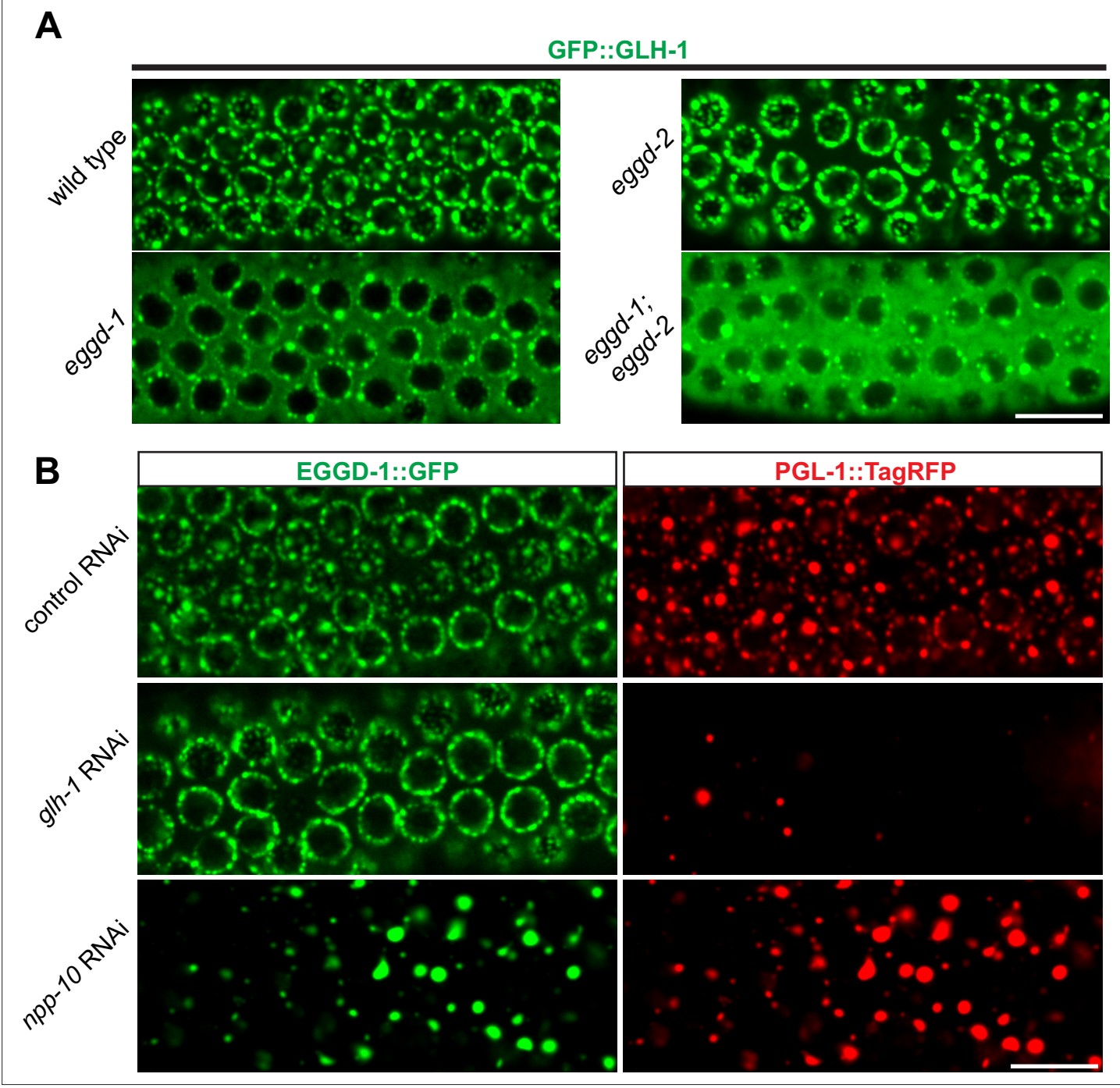

**Figure 5.** EGGD-1 acts upstream of GLH-1 in P granule assembly and localization. (**A**) Micrographs showing pachytene nuclei of wild-type and mutant animals expressing GFP::GLH-1 ( 60× objective). The contrast of images in *eggd-1,* and *eggd-1; eggd-2* mutants is manually adjusted to visualize localization of GFP::GLH-1. Images are representative of at least four animals. Scale bar=10 µm. (**B**) Fluorescence micrographs of pachytene nuclei from live animals expressing EGGD-1::GFP and PGL-1::TagRFP under the indicated RNAi conditions ( 60× objective). The contrast in images upon *glh-1* RNAi and *npp-10* RNAi is manually adjusted to visualize EGGD-1::GFP. Scale bar=10 µm.

Taken together, the epistasis analysis demonstrates perinuclear EGGD-1 requires intact nuclear pores and recruitment of GLH-1 to the nuclear periphery depends on EGGD-1.

## Ectopic expression of EGGD-1 drives formation of perinuclear granules

So far, our data suggest that EGGD-1 is necessary for the perinuclear localization of P granules. We wondered if EGGD-1 alone is sufficient to recruit GLH-1 to form perinuclear granules. To test this idea, an ectopic expression system was employed. EGGD-1 and/or GLH-1 were expressed under the muscle-specific *myo-3* promoter so that these proteins were expressed in somatic cells. This enabled us to examine the relationship between these proteins and granule assembly independently of other P granule or germline proteins.

Consistent with previous findings that wild-type GLH-1 cannot form granules by itself (*Updike et al., 2011*), mCherry::GLH-1 was diffuse throughout the cytoplasm in muscle cells (*Figure 6A*). In contrast, EGGD-1::GFP self-aggregated into granules when it was expressed ectopically. While a few cytoplasmic granules were observed, the majority of EGGD-1::GFP granules appeared to associate with the nuclear periphery (*Figure 6B*). We next drove the ectopic expression of EGGD-1::GFP and mCherry::GLH-1 simultaneously. Strikingly, both proteins co-localized and formed perinuclear foci (*Figure 6C*). Taken together, these findings indicate that EGGD-1 is intrinsically capable of forming perinuclear granules, and drives the formation of perinuclear granules by recruiting GLH-1.

## Discussion

Since the discovery of perinuclear P granules, the molecular mechanisms for their assembly have been under intensive investigation (*Strome and Wood, 1982*; *Updike and Strome, 2010*). Previous studies showed that P granule formation requires core components including PGL family members PGL-1 and PGL-3, and Vasa family members GLH-1 and GLH-4 (*Gruidl et al., 1996*; *Kawasaki et al., 2004*; *Kawasaki et al., 1998*; *Spike et al., 2008*; *Updike et al., 2011*). Zygotic P granule assembly further requires intrinsically disordered proteins MEG-3 and MEG-4 (*Smith et al., 2016*; *Wang et al., 2014*). However, it remains unclear how P granules are formed and retained at the nuclear periphery.

Using proximity labeling and genetic approaches, we discovered and characterized EGGD-1 and EGGD-2—LOTUS-domain and IDR-containing proteins. The same proteins were identified as MEG-3 interactors and referred to as MIP-1 and MIP-2 (M̲EG-3 i̲nteracting p̲rotein) respectively (*Cipriani et al., 2021*). Both studies reveal that LOTUS-domain proteins are required *for C. elegans* germline development and fertility. More importantly, both studies provide insight into the molecular function of LOTUS-domain proteins in promoting formation of perinuclear P granules. Using in vitro pull-down and yeast two-hybrid assays, Cipriani et al show that MIPs physically interact with GLH-1, a member of the Vasa family (*Cipriani et al., 2021*). Here we show that IDRs are required to anchor EGGD-1 protein to the nuclear periphery, and that the LOTUS domains are essential for perinuclear localization of P granules, likely by recruiting GLH-1. Remarkably, EGGD-1 is capable of self-assembling into perinuclear granules. Thus, EGGD-1 is not only necessary but also sufficient for recruiting GLH-1 to the nuclear periphery.

Based on these findings, we propose a model for the molecular function of EGGD proteins (*Figure 6D*). IDRs of EGGD proteins may form multivalent interactions with cytoplasmic filaments of the nuclear pore complex. It is possible that IDRs preferentially associate with phenylalanine/glycine-rich NPPs that are intrinsically disordered (*Marnik et al., 2019*; *Updike et al., 2011*). Dimerization of LOTUS domains may serve as an interface for the binding of Vasa protein GLH-1 (*Cipriani et al., 2021*; *Jeske et al., 2015*; *Jeske et al., 2017*). Association of GLH-1 with the nuclear periphery initiates the recruitment of PGL-1 and additional P granule proteins. More experiments, both in vitro and in vivo, will be required to dissect this complex PPI network.

### Interplay between EGGD-1 and EGGD-2

The interplay between EGGD-1 and EGGD-2 seems complex. In the adult germ line, EGGD-1 and EGGD-2 function partially redundantly. EGGD-1 plays a more dominant role in promoting the perinuclear localization of P granules (*Figure 3C*). In the embryos, however, both proteins are required for P granule partitioning and attachment to the nuclear periphery (*Figure 3—figure supplement 1B*).

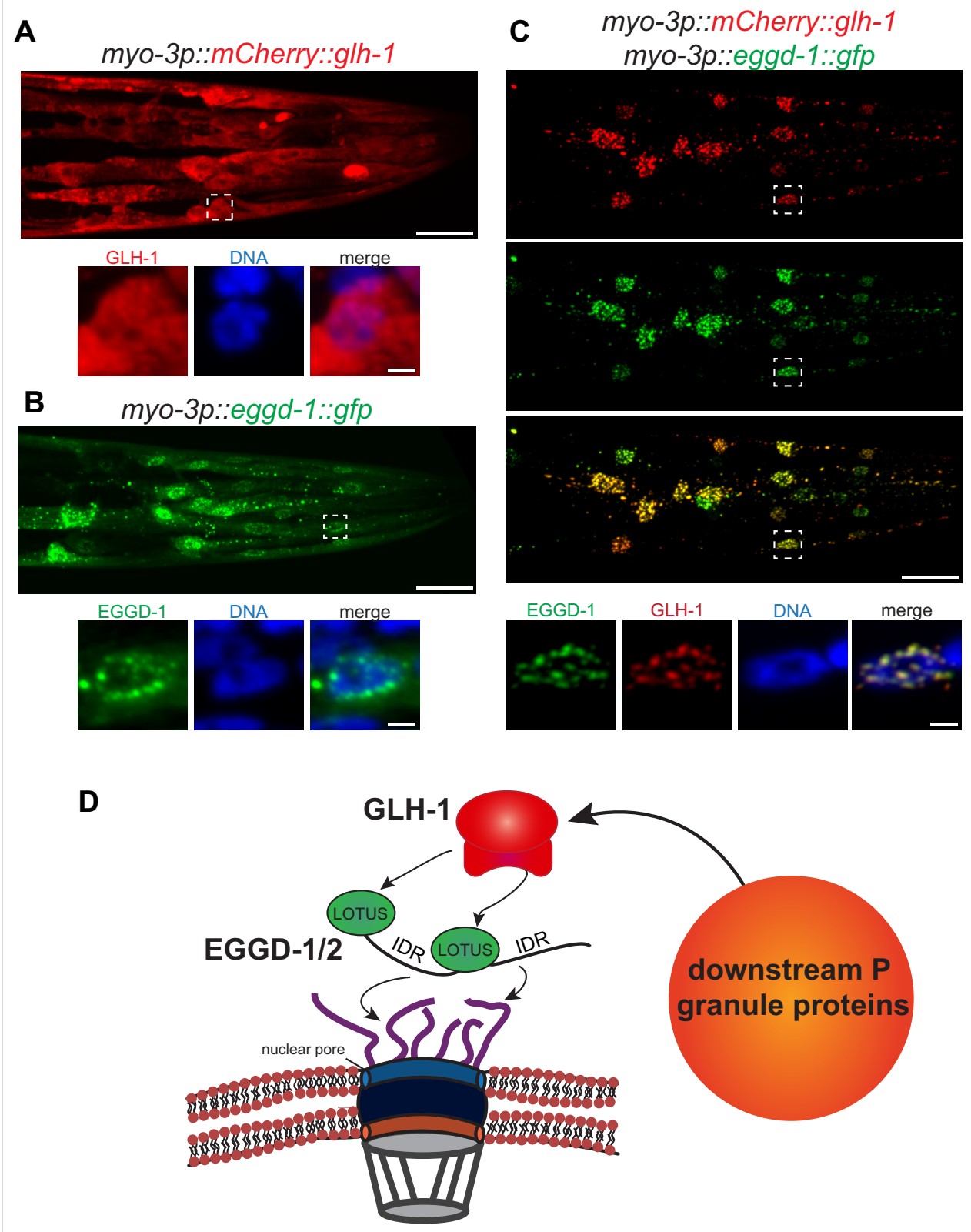

**Figure 6.** EGGD-1 intrinsically localizes to the nuclear envelope and is sufficient to recruit GLH-1 to the nuclear periphery. (**A–C**) Maximum intensity projection of a z stack spanning the head of fixed adult animals ectopically expressing mCherry::GLH-1 (**A**), EGGD-1::GFP (**B**), or mCherry::GLH-1 and EGGD-1::GFP (**C**) under the muscle-specific *myo-3* promoter. Top panel shows the entire head. Scale bar=20 μm. Bottom panel shows individual nuclei outlined by a dashed box in the top panel. Scale bar=2 μm. Images are representative of at least six animals (60× objective). (**D**) Model illustrating the proposed role of EGGD-1 in P granule assembly.

One speculative explanation is that EGGD-1 and EGGD-2 interact with different Vasa proteins and thus make distinct contributions to P granule assembly in the germ line and zygotes.

Most LOTUS domain proteins, including Oskar, TDRD5, and TDRD7, harbor a single eLOTUS domain. In contrast, EGGD-1 and EGGD-2 are predicted to contain two eLOTUS domains (*Figure 4—figure supplement 1A*). The eLOTUS domain of *Drosophilla* Oskar is capable of forming dimers (*Jeske et al., 2015*; *Jeske et al., 2017*). In vitro pull-down assays revealed that recombinant MIP-1 physically interacts with itself and MIP-2 (*Cipriani et al., 2021*). Here we show EGGD-1 self-aggregates into granules when expressed ectopically (*Figure 6*). It is possible that MIP-1/EGGD-1 and MIP-2/EGGD-2 form homodimers, heterodimers, or even oligomers in vivo through their eLOTUS domains. Of note, an independent study identified a third LOTUS domain protein LOTR-1 which is homologous to mammalian TDRD5/7 (*Marnik et al., 2021*). Interestingly, similar to EGGD-1 and EGGD-2, LOTR-1 was also enriched from our TurboID experiments (*Figure 2B* and *Supplementary file 1*). Future genetic experiments will be required to determine the interplay of these three LOTUS domain proteins in regulating perinuclear P granule formation.

## LOTUS domain proteins as scaffolds for germ granule assembly

LOTUS domains are found in bacteria, plants, and animals (*Anantharaman et al., 2010*; *Callebaut and Mornon, 2010*). LOTUS domains exhibit minimum sequence homology (*Figure 4—figure supplement 1A*; *Anantharaman et al., 2010*; *Callebaut and Mornon, 2010*). Yet they adopt a common helix-turn-helix conformation (*Anantharaman et al., 2010*; *Callebaut and Mornon, 2010*; *Jeske et al., 2015*; *Jeske et al., 2017*). The fact that diverse sequences yield a similar structure implies that the LOTUS domain acts as a structural scaffold. Indeed, the conserved function of eLOTUS domains is to bind to Vasa (*Jeske et al., 2017*).

Consistent with the idea the structure of a protein largely determines its functional properties, LOTUS proteins are essential for the development of metazoan germ cells. In *Drosophila*, Oskar is required for germ plasm assembly and germ cell formation (*Jeske et al., 2017*; *Lehmann, 2016*). In mice, TDRD7 localizes to chromatoid bodies (P granule counterpart), and is required for spermatogenesis (*Lachke et al., 2011*; *Smith et al., 2004*; *Tanaka et al., 2011*; *Yabuta et al., 2011*). In this study, we show that *C. elegans* EGGD-1 and EGGD-2 are key components for germ granule assembly on the nuclear periphery. Taken together, these findings suggest that LOTUS domain proteins belong to a unique family that has low sequence identity, but high structure homology and functional similarity. It will be important to use structure-based, but not sequence-based, search programs to identify more LOTUS domain proteins across phyla.

## Advantage and limitation of proximity labeling in this study

Proximity labeling is a powerful approach to map the proteome composition of organelles in living cells. Compared to conventional affinity purification, the key advantage lies in its ability to capture weak and transient interaction. In addition, the strong binding of biotin to streptavidin permits stringent protein extraction which reduces background contaminants. It is therefore an ideal tool to define the composition of phase-separated membraneless organelles that are formed by weak multivalent interactions (*Bracha et al., 2019*). Proximity labeling uses a promiscuous enzyme such as BioID, APEX, or TurboID (*Branon et al., 2018*; *Rhee et al., 2013*; *Roux et al., 2012*). TurboID appears to be the best choice for *C. elegans* labeling for several reasons: (1) TurboID is active from 20°C to 25°C, a range of temperature suitable for *C. elegans* cultivation (*Branon et al., 2018*). (2) TurboID uses ATP and biotin as substrates which are readily available in cells, while APEX requires exogenous cofactors which may not be easily transported into worms (*Branon et al., 2018*; *Rhee et al., 2013*). (3) TurboID catalyzes biotinylation with much greater efficiency than BioID (*Branon et al., 2018*).

In this study, we employed TurboID in combination with mass spectrometry analysis to define the constituents of P granules. While this approach identified many known and unknown P granule proteins, it has some limitations. First, TurboID fusion proteins can be toxic. We found strains expressing TurboID::IFE-1 and PGL-1::TurboID proteins either unhealthy or infertile (*Figure 1B*). Second, TurboID labeled cytoplasmic proteins as revealed by streptavidin staining (*Figure 1D*). It is possible that some proteins are biotinylated by TurboID when passing through P granules and shuttling between the nucleus and cytoplasm. Consistent with this idea, TurboID enriched IMA-2, a member of importin α family of nuclear-cytoplasmic transport factors (*Supplementary file 1*). Alternatively, because P

granules are membraneless and highly dynamic, the bait proteins (GLH-1 and DEPS-1) may constantly mix and de-mix with surrounding cytoplasmic components.

While we continue to optimize the protocol for TurboID labeling, several approaches can be used to overcome these limitations. For example, an auxin-inducible degradation system can be applied to deplete the toxic TurboID fusion proteins (*Zhang et al., 2015*). Removing worms from auxin-containing plates enables the transient expression of TurboID proteins and thus proximity labeling. More recently, a split-TurboID method was developed, in which TurboID is split into two inactive fragments, but can be reconstituted in vivo (*Cho et al., 2020*). It is conceivable to fuse individual fragments of TurboID into two P granule proteins. In this case, TurboID will become active only when two bait proteins interact within the compartment. We envision this approach will greatly improve the specificity in labeling P granule proteins.

In addition to P granules, *C. elegans* germ cells possess other membraneless organelles such as Z granules and Mutator foci (*Phillips et al., 2012*; *Wan et al., 2018*). These granules are adjacent to one another, but each has distinct functions in RNAi and transgenerational epigenetic inheritance (*Phillips et al., 2012*; *Wan et al., 2018*). We speculate that they share some common proteins, but also contain unique components. Proximity labeling described in this study can be applied to unravel the proteome of Z granules and Mutator foci, and thus will provide new insight into the organization and function of germ granules.

# Materials and methods

**Key resources table**

| Reagent type (species) or resource | Designation | Source or reference | Identifiers | Additional information |
|---|---|---|---|---|
| Strain, strain background (*Caenorhabditis elegans*) | N2 | *Caenorhabditis* Genetics Center (CGC) | N2 | Wildtype *C. elegans*, RRID:WB-STRAIN: WBStrain00000001 |
| Strain, strain background (*C. elegans*) | deps-1::TurboID | This study | WHY14 | *deps-1(how1[deps-1::TurboID]) I* |
| Strain, strain background (*C. elegans*) | TurboID::ife-1 | This study | WHY12 | *ife-1(how2[TurboID::ife-1]) III* |
| Strain, strain background (*C. elegans*) | TurboID::glh-1 | This study | WHY10 | *glh-1(how3[TurboID::glh-1]) I* |
| Strain, strain background (*C. elegans*) | pgl-1::TurboID | This study | N/A | *pgl-1(how4[pgl-1::TurboID]) IV*-- this strain is sterile and cannot be grown |
| Strain, strain background (*C. elegans*) | pgl-1::TagRFP | This study | WHY100 | *pgl-1(gg547[pgl1::3x flag::tagRFP]) IV* |
| Strain, strain background (*C. elegans*) | rrf-3; pgl-1::TagRFP | This study | WHY134 | *rrf-3(pk1426)II; pgl-1 (gg547[pgl-1::3xflag::tag RFP]) IV* |
| Strain, strain background (*C. elegans*) | eggd-1; pgl-1::TagRFP | This study | WHY219 | *eggd-1(how5) III; pgl-1(gg547[pgl-1::3x flag::tagRFP]) IV* |
| Strain, strain background (*C. elegans*) | eggd-2(17nt insertion); pgl-1::TagRFP | This study | WHY178 | *pgl-1(gg547[pgl-1::3xflag::tagRFP]) IV; eggd-2(how6) V* |
| Strain, strain background (*C. elegans*) | eggd-2(deletion);pgl-1::TagRFP | This study | WHY297 | *eggd-1(how7[eggd-1::GFP::TEV::3x FLAG::AID]) III; pgl-1(gg547[pgl-1::3x flag::tagRFP]) IV; eggd-2(how14) V* |

*Continued on next page*

*Continued*

| Reagent type (species) or resource | Designation | Source or reference | Identifiers | Additional information |
|---|---|---|---|---|
| Strain, strain background (*C. elegans*) | *eggd-1; eggd-2; pgl-1::TagRFP* | This study | WHY285 | *eggd-1(how5) III; pgl-1 (gg547[pgl-1::3xflag::tag RFP]) IV; eggd-2(how6) V* |
| Strain, strain background (*C. elegans*) | *eggd-1::GFP; pgl-1::TagRFP* | This study | WHY170 | *eggd-1(how7[eggd-1::GFP::TEV::3xFLAG::AID]) III; pgl-1(gg547[pgl-1::3x flag::tagRFP]) IV* |
| Strain, strain background (*C. elegans*) | *ΔLOTUS1* | This study | WHY203 | *eggd-1(how8[eggd-1(ΔLOTUS 1)::GFP::TEV::3x FLAG::AID]) III; pgl-1(gg547 [pgl-1::3xflag::tagRFP]) IV* |
| Strain, strain background (*C. elegans*) | *ΔLOTUS2* | This study | WHY180 | *eggd-1(how9[eggd-1(ΔLOTUS 2)::GFP::TEV::3x FLAG::AID]) III; pgl-1(gg547 [pgl-1::3xflag::tagRFP]) IV* |
| Strain, strain background (*C. elegans*) | *ΔLOTUS1; ΔLOTUS2* | This study | WHY182 | *eggd-1(how10[eggd-1(ΔLOTUS 1&2)::GFP::TEV::3x FLAG::AID]) III; pgl-1(gg547 [pgl-1::3xflag::tagRFP]) IV* |
| Strain, strain background (*C. elegans*) | *ΔIDR1* | This study | WHY186 | *eggd-1(how11[eggd-1(ΔIDR 1)::GFP::TEV::3x FLAG::AID]) III; pgl-1(gg547 [pgl-1::3xflag::tagRFP]) IV* |
| Strain, strain background (*C. elegans*) | *ΔIDR2* | This study | WHY216 | *eggd-1(how12[eggd-1(ΔIDR 2)::GFP::TEV::3x FLAG::AID]) III; pgl-1(gg 547[pgl-1::3xflag::tagRFP]) IV* |
| Strain, strain background (*C. elegans*) | *ΔIDR1; ΔIDR2* | This study | WHY282 | *eggd-1(how13[eggd-1(ΔIDR 1&2)::GFP::TEV::3x FLAG::AID]) III; pgl-1(gg547 [pgl-1::3xflag::tagRFP]) IV* |
| Strain, strain background (*C. elegans*) | *gfp::glh-1* | Gift from Craig Mello | WM704 | *glh-1(ne4816[GFP::glh-1]) I* |
| Strain, strain background (*C. elegans*) | *gfp::glh-1; eggd-1* | This study | WHY273 | *glh-1(ne4816[GFP::glh-1]) I; eggd-1(how5) III* |
| Strain, strain background (*C. elegans*) | *gfp::glh-1; eggd-2* | This study | WHY274 | *glh-1(ne4816[GFP::glh-1]) I; eggd-2(how6) V* |
| Strain, strain background (*C. elegans*) | *gfp::glh-1; eggd-1; eggd-2* | This study | WHY275 | *glh-1(ne4816[GFP::glh-1]) I; eggd-1(how5) III; eggd-2(how-6) V* |
| Strain, strain background (*C. elegans*) | *unc-119* | *Caenorhabditis* Genetics Center (CGC) | EG4322 | *ttTi5605 II; unc-119(ed9) III* |
| Strain, strain background (*C. elegans*) | *myo-3p::mCherry::glh-1* | This study | WHY276 | *ttTi5605 II; unc-119(ed9) III; howEx1[myo-3p::mCherry::glh-1::unc-54 3'UTR+ Cbr-unc-119(+)]* |
| Strain, strain background (*C. elegans*) | *myo-3p::eggd-1::GFP* | This study | WHY277 | *ttTi5605 II; unc-119(ed9) III; howEx2[myo-3p::eggd-1::GFP::unc-54 3'UTR+ Cbr-unc-119(+)]* |

*Continued on next page*

*Continued*

| Reagent type (species) or resource | Designation | Source or reference | Identifiers | Additional information |
|---|---|---|---|---|
| Strain, strain background (*C. elegans*) | *myo-3p::mCherry::glh-1; myo-3p::eggd-1::GFP* | This study | WHY278 | *ttTi5605 II; unc-119(ed9) III; howEx3[myo-3p::mCherry::glh-1::unc-54 3'UTR+ myo-3p::eggd-1::gfp::unc-54 3'UTR+ Cbr-unc-119(+)]* |
| Strain, strain background (*C. elegans*) | *GFP::csr-1* | Gift from Craig Mello | WM343 | *csr-1(GFP::csr-1) IV* |
| Strain, strain background (*C. elegans*) | *TurboID::deps-1; GFP::csr-1* | This study | WHY304 | *deps-1(how1[deps-1::TurboID]) I; csr-1 (GFP::csr-1) IV* |
| Strain, strain background (*C. elegans*) | *TurboID::deps-1; pgl-1::TagRFP* | This study | WHY305 | *deps-1(how1[deps-1::TurboID]) I; eggd-1(how7[eggd-1::GFP::TEV::3X FLAG::AID]) III; pgl-1(gg547[pgl-1::3xFLAG::TagRFP]) IV* |
| Strain, strain background (*C. elegans*) | *glh-1::TurboID; GFP::csr-1* | This study | WHY312 | *glh-1(how3[TurboID::glh-1]) I; csr-1(GFP::csr-1) IV* |
| Strain, strain background (*C. elegans*) | *glh-1::TurboID; pgl-1::TagRFP* | This study | WHY313 | *glh-1(how3[TurboID::glh-1]) I; eggd-1(how7[eggd-::GFP::TEV::3XFLAG::AID]) III; pgl-1(gg547[pgl-1::3x FLAG::TagRFP]) IV* |
| Strain, strain background (*Escherichia coli*) | OP50 | *Caenorhabditis* Genetics Center (CGC) | OP50 | Bacteria. Uracil auxotroph. *E. coli* B. |
| Strain, strain background (*E. coli*) | HT115 | *Caenorhabditis* Genetics Center (CGC) | HT115 | *E. coli [F-, mcrA, mcrB, IN(rrnD-rrnE)1, rnc14::Tn10(DE3 lysogen: lacUV5 promoter) -T7 polymerase].*, RRID:WB-STRAIN:WBStrain00041080 |
| Genetic reagent (*E. coli*) | *Control RNAi* | DOI: 10.1016/s1046-2023(03)00050–1 | L4440 | (*C. elegans* RNAi Collection (Ahringer), RRID:SCR_017064) |
| Genetic reagent (*E. coli*) | *C38D4.4* | DOI: 10.1016/s1046-2023(03)00050–1 | WBGene00008005 | (*C. elegans* RNAi Collection (Ahringer), RRID:SCR_017064) |
| Genetic reagent (*E. coli*) | *wago-1* | DOI: 10.1016/s1046-2023(03)00050–1 | WBGene00011061 | (*C. elegans* RNAi Collection (Ahringer), RRID:SCR_017064) |
| Genetic reagent (*E. coli*) | *hsp-110* | DOI: 10.1101/gr.2505604 | WBGene00016250 | Vidal RNAi Library |
| Genetic reagent (*E. coli*) | *cpf-2* | DOI: 10.1016/s1046-2023(03)00050–1 | WBGene00000774 | (*C. elegans* RNAi Collection (Ahringer), RRID:SCR_017064) |
| Genetic reagent (*E. coli*) | *puf-5* | DOI: 10.1016/s1046-2023(03)00050–1 | WBGene00004241 | (*C. elegans* RNAi Collection (Ahringer), RRID:SCR_017064) |
| Genetic reagent (*E. coli*) | *ran-2* | DOI: 10.1016/s1046-2023(03)00050–1 | WBGene00004303 | (*C. elegans* RNAi Collection (Ahringer), RRID:SCR_017064) |
| Genetic reagent (*E. coli*) | *cey-2* | DOI: 10.1016/s1046-2023(03)00050–1 | WBGene00000473 | (*C. elegans* RNAi Collection (Ahringer), RRID:SCR_017064) |

*Continued on next page*

*Continued*

| Reagent type (species) or resource | Designation | Source or reference | Identifiers | Additional information |
|---|---|---|---|---|
| Genetic reagent (*E. coli*) | cey-3 | DOI: 10.1016/s1046-2023(03)00050–1 | WBGene00000474 | (*C. elegans* RNAi Collection (Ahringer), RRID:SCR_017064) |
| Genetic reagent (*E. coli*) | ubh-4 | DOI: 10.1016/s1046-2023(03)00050–1 | WBGene00006724 | (*C. elegans* RNAi Collection (Ahringer), RRID:SCR_017064) |
| Genetic reagent (*E. coli*) | ani-1 | DOI: 10.1016/s1046-2023(03)00050–1 | WBGene00013038 | (*C. elegans* RNAi Collection (Ahringer), RRID:SCR_017064) |
| Genetic reagent (*E. coli*) | ifg-1 | DOI: 10.1016/s1046-2023(03)00050–1 | WBGene00002066 | (*C. elegans* RNAi Collection (Ahringer), RRID:SCR_017064) |
| Genetic reagent (*E. coli*) | maph-1.2 | DOI: 10.1016/s1046-2023(03)00050–1 | WBGene00009113 | (*C. elegans* RNAi Collection (Ahringer), RRID:SCR_017064) |
| Genetic reagent (*E. coli*) | F56C9.6 | DOI: 10.1016/s1046-2023(03)00050–1 | WBGene00018950 | (*C. elegans* RNAi Collection (Ahringer), RRID:SCR_017064) |
| Genetic reagent (*E. coli*) | ppw-2 | DOI: 10.1101/gr.2505604 | WBGene00004094 | Vidal RNAi Library |
| Genetic reagent (*E. coli*) | M01H9.3 | DOI: 10.1101/gr.2505604 | WBGene00019719 | Vidal RNAi Library |
| Genetic reagent (*E. coli*) | Y37E11B.10 | DOI: 10.1101/gr.2505604 | WBGene00021381 | Vidal RNAi Library |
| Genetic reagent (*E. coli*) | F01G4.4 | DOI: 10.1101/gr.2505604 | WBGene00008503 | Vidal RNAi Library |
| Genetic reagent (*E. coli*) | hmg-12 | DOI: 10.1016/s1046-2023(03)00050–1 | WBGene00001977 | (*C. elegans* RNAi Collection (Ahringer), RRID:SCR_017064) |
| Genetic reagent (*E. coli*) | edc-4 | DOI: 10.1016/s1046-2023(03)00050–1 | WBGene00021551 | (*C. elegans* RNAi Collection (Ahringer), RRID:SCR_017064) |
| Genetic reagent (*E. coli*) | ppw-1 | DOI: 10.1016/s1046-2023(03)00050–1 | WBGene00004093 | (*C. elegans* RNAi Collection (Ahringer), RRID:SCR_017064) |
| Genetic reagent (*E. coli*) | frm-4 | DOI: 10.1016/s1046-2023(03)00050–1 | WBGene00001491 | (*C. elegans* RNAi Collection (Ahringer), RRID:SCR_017064) |
| Genetic reagent (*E. coli*) | haf-9 | DOI: 10.1016/s1046-2023(03)00050–1 | WBGene00001819 | (*C. elegans* RNAi Collection (Ahringer), RRID:SCR_017064) |
| Genetic reagent (*E. coli*) | prg-1 | DOI: 10.1016/s1046-2023(03)00050–1 | WBGene00004178 | (*C. elegans* RNAi Collection (Ahringer), RRID:SCR_017064) |
| Genetic reagent (*E. coli*) | F52B5.3 | DOI: 10.1016/s1046-2023(03)00050–1 | WBGene00009922 | (*C. elegans* RNAi Collection (Ahringer), RRID:SCR_017064) |
| Genetic reagent (*E. coli*) | hip-1 | DOI: 10.1101/gr.2505604 | WBGene00010281 | Vidal RNAi Library |
| Genetic reagent (*E. coli*) | F58G11.3 | DOI: 10.1101/gr.2505604 | WBGene00008385 | Vidal RNAi Library |
| Genetic reagent (*E. coli*) | lotr-1 | DOI: 10.1101/gr.2505604 | WBGene00008399 | Vidal RNAi Library |

*Continued*

| Reagent type (species) or resource | Designation | Source or reference | Identifiers | Additional information |
|---|---|---|---|---|
| Genetic reagent (*E. coli*) | D2005.4 | DOI: 10.1016/s1046-2023(03)00050–1 | WBGene00004143 | (*C. elegans* RNAi Collection (Ahringer), RRID:SCR_017064) |
| Genetic reagent (*E. coli*) | pqn-59 | DOI: 10.1101/gr.2505604 | WBGene00002263 | Vidal RNAi Library |
| Genetic reagent (*E. coli*) | lea-1 | DOI: 10.1101/gr.2505604 | WBGene00001029 | Vidal RNAi Library |
| Genetic reagent (*E. coli*) | dnj-11 | DOI: 10.1101/gr.2505604 | WBGene00011735 | Vidal RNAi Library |
| Genetic reagent (*E. coli*) | npp-10 | DOI: 10.1016/s1046-2023(03)00050–1 | WBGene00003796 | (*C. elegans* RNAi Collection (Ahringer), RRID:SCR_017064) |
| Genetic reagent (*E. coli*) | glh-1 | doi: 10.1534/genetics.107.083469 | WBGene00001598 | |
| Recombinant DNA reagent (Plasmid) | TurboID HDR donor template | DOI: 10.1038/nbt.4201 | pAS31 | RRID:Addgene_118220 |
| Recombinant DNA reagent (Plasmid) | *pCFJ104* | DOI: 10.1038/ng.248 | *pCFJ104* | RRID:Addgene_19328 |
| Recombinant DNA reagent (Plasmid) | pCFJ151 | DOI: 10.1038/ng.248 | pCFJ151 | RRID:Addgene_19330 |
| Recombinant DNA reagent (Plasmid) | *myo-3p::eggd-1::gfp* | This study | pIP1 | myo-3p::eggd-1::gfp::unc-54 3' UTR |
| Recombinant DNA reagent (Plasmid) | *myo-3p::mCherry::glh-1* | This study | pIP12 | myo-3p::mCherry::glh-1::unc-54 3' UTR |
| Commercial assay or kit | 18×18 Cover Glass # 1 | MedSupply Partners | Cat# G07-140110 | |
| Commercial assay or kit | Microscope Slides, Diamond White Glass, 25×75 mm², 90° Ground Edges, Plain | MedSupply Partners | Cat# G07-1380-10 | |
| Commercial assay or kit | Fisherbrand Fluorescent Antibody Microscope Slides w/ two 10 mm diameter circles | Thermo Fisher Scientific | Cat# 22-339408 | |
| Commercial assay or kit | Nail Polish | Electron Microscopy Sciences | Cat# 72180 | |
| Commercial assay or kit | cOmplete, Mini EDTA-free Protease Inhibitor Cocktail | Sigma-Aldrich | Cat# 11836170001 | |
| Commercial assay or kit | Streptavidin-Alexa Fluor 488 conjugate | Life Technologies | Cat# S11223 | |
| Commercial assay or kit | Streptavidin-Horseradish Peroxidase conjugate | Life Technologies | Cat# S911 | |
| Commercial assay or kit | Streptavidin magnetic beads | NEB | Cat# S1421S | |
| Commercial assay or kit | NuPAGE 4–12% Bis-Tris Gel | Invitrogen | cat# NP0323BOX | |
| Commercial assay or kit | Immobilon-FL PVDF membrane | Sigma-Aldrich | Cat# IPFL00010 | |
| Commercial assay or kit | Vectashield antifade mounting medium with DAPI | Vector Labs | Cat# H-1200-10 | |
| Commercial assay or kit | NOVEX colloidal blue staining kit | Invitrogen | Cat# LC6025 | |
| Commercial assay or kit | Lysing Matrix D | mpbio | Cat# 6913100 | |

*Continued on next page*

*Continued*

| Reagent type (species) or resource | Designation | Source or reference | Identifiers | Additional information |
|---|---|---|---|---|
| Chemical compound, drug | Agarose | Genesee Scientific | Cat# 20-102GP | |
| Chemical compound, drug | Paraformaldehyde | Thermo Fisher Scientific | Cat# O4042-500 | |
| Chemical compound, drug | Methanol | Thermo Fisher Scientific | Cat# A408-4 | |
| Chemical compound, drug | Acetone | Thermo Fisher Scientific | Cat# A929-1 | |
| Chemical compound, drug | Sodium Chloride | VWR | Cat# BDH9286 | |
| Chemical compound, drug | Agar | Genesee Scientific | Cat# 20-248 | |
| Chemical compound, drug | Chloesterol | VWR | Cat# 0433-250 | |
| Chemical compound, drug | Magnesium Chloride | Thermo Fisher Scientific | Cat# M35-500 | |
| Chemical compound, drug | Potassium Phosphate Monobasic | Thermo Fisher Scientific | Cat# BP362-1 | |
| Chemical compound, drug | Potassium Phosphate Dibasic | Thermo Fisher Scientific | Cat# BP363-1 | |
| Chemical compound, drug | Sodium Citrate | Thermo Fisher Scientific | Cat# S297-500 | |
| Chemical compound, drug | HEPES | Sigma-Aldrich | Cat# H4034-500 | |
| Chemical compound, drug | Potassium Hydroxide | Thermo Fisher Scientific | Cat# P250-1 | |
| Chemical compound, drug | Urea | Thermo Fisher Scientific | Cat# AC140750010 | |
| Chemical compound, drug | Trition X-100 | Thermo Fisher Scientific | Cat# BP151-500 | |
| Chemical compound, drug | Tween 20 | Thermo Fisher Scientific | Cat# BP337-500 | |
| Software, algorithm | R version 4.0.3 | The R Project for Statistical Computing | https://www.r-project.org/ | RRID:SCR_001905 |
| Software, algorithm | ggplot2 version 3.3.2 | Tidyverse | https://www.tidyverse.org/ | RRID:SCR_019186 |
| Software, algorithm | ImageJ | National Institutes of Health | https://imagej.nih.gov/ij/ | RRID:SCR_003070 |
| Software, algorithm | Adobe Illustrator | Adobe | https://www.adobe.com/products/illustrator.html | RRID:SCR_010279 |
| Software, algorithm | HHPRED | DOI: 10.1016/j.jmb.2017.12.007 | https://toolkit.tuebingen.mpg.de/tools/hhpred | RRID:SCR_010276 |
| Software, algorithm | Zen Blue 3.0 | Carl Zeiss Microscopy GmbH | https://www.zeiss.com/microscopy/int/home.html | RRID:SCR_013672 |
| Software, algorithm | MetaMorph Premier Acquisition version 7.8.1.0 | Molecular Devices | https://www.moleculardevices.com/ | RRID:SCR_002368 |
| Software, algorithm | g:Profiler | doi:10.1093/nar/gkz369 | https://biit.cs.ut.ee/gprofiler/gost | RRID:SCR_006809 |

*Continued on next page*

*Continued*

| Reagent type (species) or resource | Designation | Source or reference | Identifiers | Additional information |
|---|---|---|---|---|
| Software, algorithm | IUPred2A | DOI: 10.1093/bioinformatics/bti541 | https://iupred2a.elte.hu/ | RRID:SCR_014632 |
| Software, algorithm | STRING | DOI: 10.1093/nar/gki005 | https://string-db.org/ | RRID:SCR_005223 |
| Software, algorithm | PROMALS3D | doi: 10.1093/nar/gkn072 | http://prodata.swmed.edu/promals3d/promals3d.php | RRID:SCR_018161 |
| Peptide, recombinant protein | Alt-R S.p. Cas9 Nuclease V3 Cat# 1081058 | Integrated DNA Technologies | Cat# 1081058 | |
| Commercial assay or kit | Alt-R CRISPR-Cas9 tracrRNA, 20 nmol | Integrated DNA Technologies | Cat# 1072533 | |
| Sequence-based reagent | ife-1 5' guide RNA | Integrated DNA Technologies | Guide RNA oligo | TTGAGAAGCTGAAAATCTCT |
| Sequence-based reagent | deps-1 3' guide RNA | Integrated DNA Technologies | Guide RNA oligo | gtatatatttaaTTAGACCC |
| Sequence-based reagent | glh-1 5' guide RNA | Integrated DNA Technologies | Guide RNA oligo | ttttctgcgaaaATGTCTGA |
| Sequence-based reagent | pgl-1 3' guide RNA | Integrated DNA Technologies | Guide RNA oligo | tagaaattattaaaggcgcA |
| Sequence-based reagent | eggd-1 5' guide RNA | Integrated DNA Technologies | Guide RNA oligo | GACATTCACTTGGCAAATGA |
| Sequence-based reagent | eggd-1 3' guide RNA | Integrated DNA Technologies | Guide RNA oligo | CACCAACTATCCTTATCCGA |
| Sequence-based reagent | eggd-2 5' guide RNA | Integrated DNA Technologies | Guide RNA oligo | TGAAAAATGTCTGAAGAAGA |
| Sequence-based reagent | eggd-2 3' guide RNA | Integrated DNA Technologies | Guide RNA oligo | GCACTGCTTCAACTACGCCT |
| Sequence-based reagent | eggd-1 5' ΔLOTUS1 guide RNA | Integrated DNA Technologies | Guide RNA oligo | CGACCCCAAATCAAGTAGAA |
| Sequence-based reagent | eggd-1 3' ΔLOTUS1 guide RNA, 5' ΔIDR1 guide RNA | Integrated DNA Technologies | Guide RNA oligo | GCTTTGAGATCAGATTGATT |
| Sequence-based reagent | eggd-1 5' ΔLOTUS2 guide RNA, 3' ΔIDR1 guide RNA | Integrated DNA Technologies | Guide RNA oligo | TGGCTGCAACTCGGAACAGA |
| Sequence-based reagent | eggd-1 3' ΔLOTUS2 guide RNA | Integrated DNA Technologies | Guide RNA oligo | ATACACTTCGAGTCAATCCC |
| Sequence-based reagent | eggd-1 5' ΔIDR2 guide RNA | Integrated DNA Technologies | Guide RNA oligo | GGAACTCCAAAAGATCTTCC |
| Sequence-based reagent | eggd-1 3' ΔIDR2 guide RNA | Integrated DNA Technologies | Guide RNA oligo | CTCCAGCTGTCTTTGTCTGA |
| Sequence-based reagent | turboID::ife-1 5' homology arm | Integrated DNA Technologies | 5' primer to amplify dsDNA donor | cacgattagttggcgttttccccagttgtt ctcggcttctcagatcagtcctgtttttgcc ttgccagttgtcgaggtgc gaaaatttta agcgcaaATG tacccatacga CgtCccaga |
| Sequence-based reagent | turboID::ife-1 3' homology arm | Integrated DNA Technologies | 3' primer to amplify dsDNA donor | tgaataatttatagtactcaaacga taatgaaaaagggaatggctcac CTTCTTTCTCTCCAGAG ATTTTCAGCTTCTCAAAT GCTATTTCAGAATCTGA CTTCTCGGCGGAACGAAGGG |

*Continued*

| Reagent type (species) or resource | Designation | Source or reference | Identifiers | Additional information |
|---|---|---|---|---|
| Sequence-based reagent | deps-1::turboID 5' homology arm | Integrated DNA Technologies | 5' primer to amplify dsDNA donor | CAGTGAGCTCAAACgtaag tttattttaaggttggaagatgataaaa acaagttttcagCGATTCGTT GGCCCTTCAA GCCGCA GAACTCCATCTGGTACTC CACAAAGCTCAACATCTT CCAGGGTC tacccatacga CgtCccaga |
| Sequence-based reagent | deps-1::turboID 3' homology arm | Integrated DNA Technologies | 3' primer to amplify dsDNA donor | gaatgggatggtggtggaacttga agtttaaataaataaatgtttggttg gataacgggtagattaaaaatga gcagaacatttgaaacacaaat acggggaaaacgggatgcgt atatatttaaTTA CTTCTCG GCGGAACGAAGGG |
| Sequence-based reagent | turboID::glh-1 5' homology arm | Integrated DNA Technologies | 5' primer to amplify dsDNA donor | acctcgacacactcatctacta aattttgggacagttcctaattctt tttgctgttttcaactcaattttctg gaaaaat cttaattttctgcgaa aATG tacccatacga CgtCccaga |
| Sequence-based reagent | turboID::glh-1 3' homology arm | Integrated DNA Technologies | 3' primer to amplify dsDNA donor | CTACCGAATCCAGT TTTGGctgaaataaagtttt taatcaaaataaaaccggtgg aaagttcaaaataaaactcac CCTTAGCAGCACTTT CGCTATCACTCCAAC CATCAGA CTTCTCG GCGGAACGAAGGG |
| Sequence-based reagent | pgl-1::turboID 5' homology arm | Integrated DNA Technologies | 5' primer to amplify dsDNA donor | tctataaaatctataacaagt taaacatattatttaattataa aaccccgcattgattaaacat attttgatttgaaaaa aaaaac tagaaaataggtaaaataaatc tggaaatagttcagaaac ttagaaattattaaaggcgc ATGtacccatacgaCgtCccaga |
| Sequence-based reagent | pgl-1::turboID 3' homology arm | Integrated DNA Technologies | 3' primer to amplify dsDNA donor | ttcgagattagaattcaaaaa aacgcaaaatttacCCAAA AAAGTAAGAAAACGG AAAAGAAAATTGGG ACGAGATCGAAA TTGCAACTTCCG CGTTCGCGTCGAG TTGTTCGTTTCGAGA CCCGTAGATC TGAAACTTC CTTCTCGGCG GAACGAAGGG |
| Sequence-based reagent | eggd-1::FLAG::AID::GFP::TEV 5' homology arm | Integrated DNA Technologies | 5' primer to amplify dsDNA donor | TACAAAAGTGCCAT CCACGACTAGAAG TGTAGTTCTCCCAC CAATGTCAAAAGGA CCAGGATTGGCAC GTTCTCGTAACT TTTCACCACAACAA TCGACTACATCTTCA ATTGATAATGAGTGT CTAGAAGCTATCAAT GCTGCGTTGCCG TCaGAcAAa GAcAGc TGGAGATCCAGTAA AGGAGAAGA |

*Continued on next page*

*Continued*

| Reagent type (species) or resource | Designation | Source or reference | Identifiers | Additional information |
|---|---|---|---|---|
| Sequence-based reagent | eggd-1::FLAG::AID::GFP::TEV 5' homology arm | Integrated DNA Technologies | 3' primer to amplify dsDNA donor | tgaatgactcgcatccaaa atataaaaaaacaatgtt actattaaaactaattaaa aaataattttacaaaaac acata aacaggatatttt aaagcacgtaaaatttcga TCActtcacgaacgccgccgcct |
| Sequence-based reagent | mCherry::eggd-2 5' homology arm | Integrated DNA Technologies | 5' primer to amplify dsDNA donor | acttctgccacgattttgac atttttaagtttaaatcatttt tttgtattcgttatttcagatt tccgtttctgaata tttaa agtcattcaactgattgttttac tgtttccagcatttgcctgaaaa ATGGTCTCAAAGG GTGAAGAAGA |
| Sequence-based reagent | mCherry::eggd-2 3' homology arm | Integrated DNA Technologies | 3' primer to amplify dsDNA donor | AAGAAGAACACTATA AGCGTCCCGTTCGA TGCGCTTACGCATTT TGTTCATTTTTTCTTT GCCcTCcTCcTCtGAA GCTC CACCTCCACCTCCCTT |
| Sequence-based reagent | ΔLOTUS1 single strand donor | Integrated DNA Technologies | Single-stranded repair template | AGAAAATACGCGCG CGCATCGAGCGCGA CGTGTACAGTGTTCT GCTATCAAAGAAAAAA AAGAAAGGTGGAAA AGGTGCA AAGCCCA TTCGTGCAGCTAC |
| Sequence-based reagent | ΔLOTUS2 single-strand donor | Integrated DNA Technologies | Single-stranded repair template | TTGTTCAAAGACTGT CGTCTACAGTGGCTGT TCCAGTTTTGCAACCCG GGAAGGAACCCTGGTTC ACAAATTTTGGAGCT GC GTTAAAGAAATCAATGCC |
| Sequence-based reagent | ΔIDR1 single strand donor | Integrated DNA Technologies | Single-stranded repair template | caatacattcgttttcagCGCA AATGAGGACAATCA AGAtCTAATtAAcCTtA TtTCtAAACAaAAaA AgAAgAAaAAgG GaGGAAAgGGaACaG TcGTgCAgAGACTtTC tTCaACtGTtGCaGTgC CgGTcTTaCAgCCaG GaATcGAtTCaAAaTGc ATGCCTTCGATCGTT GATTTTTCGAACAA CGTTAAGCGCA |
| Sequence-based reagent | ΔIDR2 single strand donor | Integrated DNA Technologies | Single-stranded repair template | ttttagGACTCACAGTG AGTGCCCGTAGCGTA ATGAGATCCAGTAAA GGAGAAGAACTTT TCACTGGAGT |
| Sequence-based reagent | 5' amplify eggd-1::GFP for cloning | Integrated DNA Technologies | PCR primer | CTTCTT CCTAGG ATG ACGGAAGCTGA CGATCCCAA |
| Sequence-based reagent | 3' amplify eggd-1::GFP for cloning | Integrated DNA Technologies | PCR primer | CTTCTT GAGCTC TCA CGATTGG AAGTAGAGGTTCT |
| Sequence-based reagent | 5' amplify mCherry::GLH-1 for cloning | Integrated DNA Technologies | PCR primer | CTTCTT CCTAGG ATGGTCTCAAAGG GTGAAGAAGATAACATG |
| Sequence-based reagent | 3' amplify mCherry::GLH-1 for cloning | Integrated DNA Technologies | PCR primer | CTTCTT GAGCTC CTACCAGCCTT CTTCATCTTGA |

## Strains

Worms were cultured according to standard methods at 20°C unless otherwise indicated (*Brenner, 1974*). N2 strain serves as untagged control. *pgl-1::TagRFP* strain serves as wild-type in germ line atrophy characterization, brood size counting, and germ line mortality assay. Mutant animals were generated using CRISPR editing or obtained from the CGC. All strains used in this study are listed in the Key resources table.

## CRISPR genome editing

TurboID strains, *eggd-1::GFP::TEV::FLAG::AID,* and *mCherry::eggd-2* were generated by the co-CRISPR strategy (*Kim et al., 2014*). A vector containing a dominant allele of *rol-6* was used as a co-injection marker. Repair templates for insertion were made by PCR and purified by agarose gel purification. Prior to injection, repair templates were melted to improve HDR efficiency (*Ghanta and Mello, 2020*). Roller F1 were picked and presence of insertions at edited loci were screened for by PCR. Domain deletion alleles of *eggd-1* were generated by co-injecting guide RNAs targeting the region flanking the deletion. Single-stranded DNA donors served as a repair template for precise mutations as described (*Paix et al., 2017*). Guide RNA sequences, repair oligo sequences, and primer sequences for repair template amplification can be found in the Key resources table.

## Molecular cloning and generation of ectopic expression strains

Endogenously tagged loci of *eggd-1::GFP*, and *mCherry::glh-1* were amplified by PCR. pCFJ104 (Pmyo-3::mCherry::unc-54) was digested with XbaI and SacI, the plasmid backbone was separated by gel purification. PCR products were digested with XbaI or AvrII and SacI and ligated into the pCFJ104 backbone. Plasmids were isolated by miniprep and sequence-verified by sanger sequencing. *unc-119(ed9)* animals were injected with a plasmid mixture including the genes of interest and pCFJ151 that contains C.Briggsae unc-119 rescue gene (*Frøkjaer-Jensen et al., 2008*). non-unc F1 progeny were single-picked. non-unc F2 progeny were used to establish at least three independent transmission lines that carry extrachromosomal arrays.

## RNAi by feeding

The HT115 RNAi feeding strains were picked from the *C. elegans* RNAi Collections (Key resources table) (*Kamath and Ahringer, 2003*; *Lamesch et al., 2004*). All RNAi experiments were performed on NGM plates supplemented with 50 µg/ml ampicillin and 5 mM IPTG. For the genetic analysis, L4 larvae were transferred to plates seeded with HT115 bacteria expressing dsRNAs against the gene of interest. After 4–5 days, their progeny were imaged and scored.

## Brood size counting

TurboID animals were assayed at 15°C. Wild-type, *eggd-1*, *eggd-2*, and *eggd-1; eggd-2* animals were assayed at 20°C at approximately generation 12 after outcrossing one time to wild-type. Newly hatched L1 larvae were placed singly on plates. Halfway through egg-laying, animals were transferred to fresh plates until egg-laying stopped. The brood size for each animal was calculated by adding the progeny on the original and transferred plates.

## Germ line mortality assay

Prior to starting the assay, animals were freshly outcrossed once to wild-type animals. 10 lines of wild-type, *eggd-1*, *eggd-2*, and *eggd-1; eggd-2* were grown continuously at 20°C on OP50. Four L1 larvae for each line were transferred to fresh plates seeded with OP50 bacteria every two generations (approximately every 4–5 days). Lines were recorded as fertile until animals no longer produce viable progeny.

## Streptavidin staining

Animals were synchronized by hypochlorite lysis. Synchronized L1 were transferred to NGM seeded with OP50 and incubated at 15°C until L4 stage. Animals were then transferred to 25°C overnight for biotin labeling. Worms were suspended in M9 and washed three times. Animals were then transferred to M9 supplemented with 0.25 mM levamisole and dissected on slides coated with poly-L-Lysine. Slides were submerged in – 20°C methanol for 10 min, followed by 15 min in 3% paraformaldehyde

solution in phosphate-buffered saline (PBS) at room temperature. Paraformaldehyde solution was removed, and slides were washed 3× in PBS and then 1:2000 streptavidin-Alexa Fluor 488 in PBS + 0.1% Triton X-100 was added to the slides. Slides were kept in a humidified chamber overnight at 4°C. The slides were then washed four times for 30 min with PBS + 0.1% Triton X-100 and two times with PBS. 5 µl of antifade media with DAPI was added to the slides and covered with a coverslip. The slides were sealed with nail polish and kept at 4°C until imaging.

### Whole worm fixation

Animals carrying *myo-3p::glh-1* and/or *myo-3p::eggd-1* extrachromosomal arrays were resuspended in M9 and washed two times to remove bacteria. M9 was replaced with – 20°C methanol. Animals were incubated at – 20°C for 15 min and then briefly centrifuged at 100×*g*. methanol was removed and replaced with – 20°C acetone. Tubes were kept at – 20°C for 20 min, and animals were centrifuged as before. Acetone was replaced with a solution of 50% acetone, 50% ultrapure water, and animals were incubated at – 20°C for 15 min. This process was repeated with 25% acetone at 4°C. Fixed animals were centrifuged and supernatant was removed. 30 µl of antifade medium+ DAPI was added to the tube. Fixed animals were transferred to slides using a glass Pasteur pipette, covered with a coverslip, and sealed with nail polish.

### Microscopy

Live animals were suspended in M9 buffer, immobilized using 0.5 mM levamisole, and mounted on fresh 5% agar pads. Live embryos were dissected from gravid adult worms in M9 buffer and transferred to agar pads for imaging. Spinning disc confocal images were acquired using a Nikon TiE inverted microscope equipped with an Andor Revolution WD spinning disc system. Images were taken using a CFI Plan Apo VC 60×/1.2NA water immersion objective, or a CFI Plan Apo VC 100×/1.4NA oil immersion objective with an Andor Neo sCMOS detector. Airyscan images were acquired using a Zeiss Axio Observer microscope equipped with an Airyscan two detector and a Plan Apo 63×/1.4NA objective. Image processing was performed using standard 3D Airyscan processing.

### Quantification of PGL-1::TagRFP in the rachis and edge of the germ line

Single-plane 54×37.8 µm$^2$ rectangular optical cross-section images of the adult germ line in the pachytene region were obtained for four separate animals. Due to the heterogeneity of PGL-1::tagRFP foci in *eggd-1* and *eggd-1; eggd-2* mutants, it was challenging to accurately quantify PGL-1::tagRFP signals that are associated with germ cell nuclei or rachis. Instead, we took an unbiased approach. ROI (region of interest) was drawn about each edge of the germ line, and the rachis (*Figure 3—figure supplement 1B*). The background fluorescence of the image was determined by drawing an ROI outside of where the animal was located. The mean intensity of each ROI was determined using the measure function in ImageJ and the ratio of rachis and edge PGL-1::TagRFP was calculated as follows:

$$\frac{2*\left(rachis-background\right)}{\left(edge1+edge2\right)-\left(2*background\right)}$$

### Germ line atrophy characterization

Wild-type animals and *eggd* mutants expressing PGL-1::TagRFP were synchronized by hypochlorite lysis and plated to NGM on day 1. Animals were maintained at 20°C until day 4 and imaged using a Leica DMIRE2 inverted microscope equipped with a Zeiss Axiocam 305 mono camera using a dry 40× objective. Germ line defects were broadly sorted into four categories: normal, small, atrophy, and absent/few cells.

### Streptavidin-HRP blotting

100 synchronized L1 animals were plated to NGM. Animals were grown at 15°C until they reached the L4 stage. Animals were then incubated overnight at 25°C and harvested. Worm lysates were prepared by boiling animals at 100°C, and then were separated on precast denaturing polyacrylamide gels, transferred onto PVDF membrane (Bio-Rad), and probed with HRP-Conjugated Streptavidin 1:4000 (Thermo Fisher Scientific) for detection using Clarity ECL Western Blotting Substrate (Bio-Rad). The blot then was stained by Coomassie blue (Thermo Fisher Scientific).

## TurboID proximity-based labeling

TurboID strains and N2 animals were synchronized by hypochlorite lysis. 40,000 synchronized L1 animals were plated to NGM seeded with concentrated OP50 food. Animals were grown at 15°C until they reached the L4 stage. Animals were then incubated overnight at 25°C, collected in M9 and washed two times in M9, once in ddH$_2$O and once in RIPA buffer (50 mM Tris-HCl (pH 7.5), 150 mM NaCl, 0.125% SDS, 0.125% sodium deoxycholate, 1% Triton X-100 in ddH$_2$O). Animals were then resuspended in RIPA buffer supplemented with cOmplete mini EDTA-free Protease Inhibitor Cocktail tablets (Sigma-Aldrich). Resuspended pellets of animals were flash-frozen in liquid N$_2$ until further use. Worm pellets were lysed using a bead mill homogenizer. Lysate was centrifuged at 14,000× RPM. The supernatant was mixed with 80 μl Streptavidin magnetic beads (Thermo Fisher Scientific) and incubated overnight at 4°C with constant rotation. Beads were then washed for 5 min, two times with RIPA buffer, once with 1 M KCl, once with 0.1 M Na$_2$CO$_3$, and once with 2 M urea in 10 mM Tris-HCl (pH 8.0). Beads were resuspended in PBS and subjected to on-beads trypsin digestion.

## On beads digestion and mass spectrometry analysis

Streptavidin magnetic beads were washed with 50 mM ammonium bicarbonate three times. After the third wash, DTT (Thermo Fisher Scientific) was added and the sample was incubated at 4°C for 15 min. After the incubation, iodoacetamide was added and the sample was kept in dark at room temperature for 30 min. 250 ng of sequencing grade-modified trypsin (Promega) prepared in 50 mM ammonium bicarbonate was mixed with sample at 37°C overnight. The reaction was quenched by adding acetic acid for acidification. Supernatant was taken out and concentrated for LC/MSMS analysis.

Capillary-liquid chromatography-nanospray tandem mass spectrometry of protein identification was performed on an orbitrap Fusion mass spectrometer equipped with an EASY-Spray source (Thermo Fisher Scientific). Sequence information from the MS/MS data was processed by converting the raw files into a merged file (.mgf) using MSConvert (ProteoWizard). Isotope distributions for the precursor ions of the MS/MS spectra were deconvoluted to obtain the charge states and monoisotopic $m/z$ values of the precursor ions during the data conversion. The resulting.mgf files were searched using Mascot Daemon by Matrix Science version 2.5.1 and the database was searched against *C. elegans* Uniprot database.

## TurboID protein enrichment analysis

Following pre-processing, the data were further analyzed using custom R scripts. Briefly, the mean spectral count (n=3) was compared between TurboID tagged strains and wild-type using a pseudo-count of 0.01 to account for zeros in the data; p-values were derived using one-tailed t-test. Volcano plots were generated using ggplot2 and Venn diagrams of enriched proteins were generated using BioVenn.

## Protein domain identification

Putative LOTUS domains of EGGD-1 and EGGD-2 were first identified using HHpred (*Zimmermann et al., 2018*). To confirm the presence of LOTUS domains in EGGD-1 and EGGD-2, secondary structure prediction was performed using 103 amino acid long windows about the putative LOTUS domains along with the known extended LOTUS domains from *Drosophila melanogaster* Oskar, and *Mus musculus* TDRD5 and TDRD7 using PROMALS3D (*Pei et al., 2008*). EGGD-1 and EGGD-2 LOTUS domains were determined to be extended LOTUS domains by the presence of an alpha-helical C-terminal extension (α5) (*Jeske et al., 2017*). IDRs were defined using the IUPRED server (*Dosztányi et al., 2005*). Regions with sustained IUPRED scores over 0.5 were classified as disordered regions.

## Protein-protein interaction network

Proteins enriched in both DEPS-1 and GLH-1 TurboID pull-down experiments were submitted for PPI network analysis to STRING (string-db.org) using a high confidence and full interaction method with all active interaction sources selected (*Jensen et al., 2009*). The resulting network was exported to Cytoscape for visualization (*Shannon et al., 2003*).

## Protein disorder analysis

Using a custom shell script, the longest protein isoform for each protein in the *C. elegans* proteome was parsed from a protein annotation fasta file (*C. elegans* release WS230). The per-residue disorder for each protein was then calculated using IUPRED (*Dosztányi et al., 2005*). A custom python script was then used to calculate the mean disorder for each protein by dividing total IUPRED score by protein length. A custom R script was used to compare the mean disorder of proteins between proteins enriched in either/both turboID tagged strain with that in a random control. The random sample of proteins used for comparison was generated using the sample_n() R function from the dplyr R package. A Wilcoxon rank-sum test was used to statistically compare the disorder of tagged proteins with that of the randomized control group.

## Gene ontology analysis

Enriched proteins present in both turboID tagged strains were used in GO analysis using g:Profiler (*Raudvere et al., 2019*). Results from g:Profiler GO analysis were plotted using a custom R script.

## Data availability

The full data set including raw data and metadata files is deposited to Dryad (https://doi.org/10.5061/dryad.q2bvq83k9). All scripts used in this report are available at GitHub (https://github.com/benpastore/TurboID; *Pastore, 2021*; copy archived at swh:1:rev:24fd901d6cb08c2689d08cd4d7c09173a-90decb9). Raw mass spectrometry data are available via PRIDE and ProteomeXchange under the accession number PXD027998.

## Acknowledgements

The authors thank D Schoenberg for discussion and comments; L Zhang for Mass spectrometry analysis, OSU Proteomics core for equipment (S10OD018056); A Brown at OSU Neuroscience Imaging Core for advice and instruments (S10OD010383, S10OD026842, and P30NS104177); Ohio Supercomputer Center for computing resources; C Mello for providing *gfp::glh-1* and *gfp::csr-1* strains; D Updike for providing *glh-1* RNAi construct; K McDougal for initial analyses of TurboID data. Some of the *C. elegans* strains were provided by the *Caenorhabditis* Genetics Center supported by NIH (P40-OD010440). This work was supported by NIH Pathway to Independence Award (R00GM124460) and Maximizing Investigators' Research Award (R35GM142580) to WT.

## Additional information

### Funding

| Funder | Grant reference number | Author |
| --- | --- | --- |
| National Institutes of Health | R00GM124460 | Wen Tang |
| National Institutes of Health | R35GM142580 | Wen Tang |

The funders had no role in study design, data collection and interpretation, or the decision to submit the work for publication.

### Author contributions

Ian F Price, Conceptualization, Data curation, Investigation, Methodology, Visualization, Writing - review and editing; Hannah L Hertz, Data curation, Investigation, Writing - review and editing; Benjamin Pastore, Data curation, Methodology, Visualization; Jillian Wagner, Investigation; Wen Tang, Conceptualization, Funding acquisition, Methodology, Supervision, Writing - original draft

### Author ORCIDs

Ian F Price http://orcid.org/0000-0002-7820-8958
Wen Tang http://orcid.org/0000-0001-6684-5258

Decision letter and Author response
Decision letter https://doi.org/10.7554/eLife.72276.sa1
Author response https://doi.org/10.7554/eLife.72276.sa2

## Additional files

### Supplementary files

- Supplementary file 1. P granule proteins revealed by TurboID and their IUPRED score.
- Supplementary file 2. Known components of P granules, Z granules and Mutator foci in hermaphrodite germ line.
- Supplementary file 3. Nuclear pore complex proteins that are enriched by proximity labeling.
- Supplementary file 4. Significantly enriched gene ontology terms for TurboID hits.
- Transparent reporting form

### Data availability

All data generated or analysed during this study are included in the manuscript and supporting file. Source data files are uploaded to Dryad: https://doi.org/10.5061/dryad.q2bvq83k9. Scripts for data analysis are uploaded to Github https://github.com/benpastore/TurboID copy archived at https://archive.softwareheritage.org/swh:1:rev:24fd901d6cb08c2689d08cd4d7c09173a90decb9.

The following dataset was generated:

| Author(s) | Year | Dataset title | Dataset URL | Database and Identifier |
| --- | --- | --- | --- | --- |
| Tang W | 2021 | Data from: Proximity labeling identifies LOTUS domain proteins that promote the formation of perinuclear germ granules in C. elegans | https://doi.org/10.5061/dryad.q2bvq83k9 | Dryad Digital Repository, 10.5061/dryad.q2bvq83k9 |
| Pastore B | 2021 | TurboID Mass Spectrometry Analysis | https://archive.softwareheritage.org/swh:1:dir:1c6dc7f173a31fbd28ef1939394c4915d900cfd1 | TurboID, github |

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
