## [Editor Report]

The authors use proximately labeling and genetic experiments to identify and functionally characterize new components of *C. elegans* P granules. The conclusions of the paper are well-supported by the data. This work will be of broad interest to developmental biologists, particularly those interested in the formation and function of germ cells.

---

## [Decision Letter]

**Decision letter after peer review:**

Thank you for submitting your article "Proximity labeling identifies LOTUS domain proteins that promote the formation of perinuclear germ granules in *C. elegans*" for consideration by *eLife*. Your article has been reviewed by 3 peer reviewers, including Michael Buszczak as Reviewing Editor and Reviewer #1, and the evaluation has been overseen by Piali Sengupta as the Senior Editor. The following individual involved in review of your submission has agreed to reveal their identity: Judith Yanowitz (Reviewer #3).

All three reviewers agree that this is a strong submission. The authors need to address a few points before the paper is accepted. These points are listed below. I have included the full reviews that list additional points that would further improve the paper.

1) The naming of the genes should be approved by WormBase and consistent with the other manuscript describing these same proteins (now published in *eLife*).

2) All of the TURBO-ID strains have reduced viability. Since there is some concern that P granule composition could be affected in the tagged strains, showing the localization of other known components of P granules (GLH-1, PGL-1) in these lines is critical.

3) The wording of "cytoplasmic vs perinuclear" P granule quantification a bit inaccurate – this should be modified.

4) The authors should further discuss how many of the proteins identified by TurboID were previously identified and known components of P granules, nuclear pores, or other neighboring germ granule compartments.

*Reviewer #1 (Recommendations for the authors):*

As stated above, I think this is a strong paper. No new experiments are needed in my opinion.

*Reviewer #3 (Recommendations for the authors):*

It would be important to know how many outcrosses were done of the strains.

It looks like some proteins were tagged N' terminal and some C' terminal but this is not reflected in the methods (were the C-terminal fusions really FLAG::AID::GFP::TEV? Or were they TEV::GFP::AID::FLAG?)

In the discussion about the different types of granules in the germ line and potential overlap, it would be helpful to mention how many known components of these other granules were found in their studies.

Scale bars should be provided on all images.

The naming of the gene should be approved by the WormBase and consistent with the other manuscript (now published in *eLife*).

In Figure 2B legend: Strain names should be italicized. Check the entire manuscript for correct strain designation and nomenclature.

Figure 2B: purple text is hard to distinguish from black.

Figure 2C: It is not clear if these the only core and filament proteins? If not, list all of the proteins and highlight those detected in the pull-down.

Figure 3A: Cannot understand what Figure 3A is from the legend. Instead of "gene name" in the table, how about "RNAi target"?

Figure 3B: If plg-1::turboID is sterile, can you confirm that pgl-1::TagRFP strain is not compromised by showing brood size.

Germ line is the noun form; germline is the adjective form

Figure S3C: It is unclear to me why in the images of the S3C that they is gut fluorescence in the last (few cells) category but not the other categories. Is this real signal? Also, in the atrophy image it appears that germline tissue does have a bend. Seeing these with Hoecst co-stain would be helpful to evaluate the full morphology. Also it is appears that in small germ lines, the loss is in the early germ cells and that there are cellularizing oocytes earlier in the gonad (based on nuclear size).

Figure 6 legend is showing the head not the pharynx.

Figure 6A. The inset is labelled "DAPI" but the DAPI separation is not presented to prove that is a nucleus and this is an issue as there are other cells in which the nuclei appear void of the mCherry protein.

Line 277: period missing at end of paragraph.

---

## [Author Response]

All three reviewers agree that this is a strong submission. The authors need to address a few points before the paper is accepted. These points are listed below. I have included the full reviews that list additional points that would further improve the paper.

We appreciate reviewers’ appreciative and constructive comments. We are delighted to have reviewers’ general endorsement. In response to the reviewers’ suggestions, we have revised our manuscript.

First and foremost, we have to report a corrigendum in the original article: the stock of TurboID::glh-1 was erroneously switched with TurboID::ife-1. We found it out when conducting genetic crosses. We performed three sets of experiments to confirm the switch of these two strains.

1) We conducted genotyping using glh-1 and ife-1 specific primers. PCR product representing TurboID::ife-1 sequences was found using TurboID::glh-1 gDNA as the template. PCR product representing TurboID::glh-1 sequences was found using TurboID::ife-1 gDNA as the template.

2) Because TurboID ligase contains a 3xHA epitope tag (Branon et al., 2018), we conducted Western blotting using anti-HA antibody. A band corresponding to TurboID::IFE-1 protein (64.5kDa) was detected using TurboID::glh-1 lysate, while a band corresponding to TurboID::GLH-1 protein (117.9kDa) was detected in TurboID::ife-1 lysate.

3) We examined mass spectrometry data. An unique TurboID::IFE-1 peptide “SAEKSDSEIAFEK” was found in the GLH-1 pulldown, but not the IFE-1 pulldown. In addition, an unique TurboID::GLH-1 peptide “SAEKSDGWSDSESAAK” was found in found in the IFE-1 pulldown, but not GLH-1 pulldown.

**Author response image 1. sa2fig1:** 

We have corrected this error in the revised manuscript. Conclusions of the article are not affected. However, we sincerely apologize to the reviewers and editors for this mistake. If needed, we will provide raw data and lab notebook record.

1) The naming of the genes should be approved by WormBase and consistent with the other manuscript describing these same proteins (now published in eLife).

Wormbase has approved EGGD-1 and EGGD-2 as “alias/other names” for C38D4.4 and F58G11.3, respectively. Furthermore, Wormbase will add a new gene class “eggd” for “Embryonic and Germline P Granule Detached”. All updates will be available in the upcoming WS283 release.

We prefer to use EGGD for three reasons: (1). These two genes were independently identified using a proximity labeling approach in which MEG-3 was not the bait; (2). Our manuscript does not investigate the interaction between EGGD proteins and MEG-3; (3). The name reflects on the mutant phenotype. We hope reviewers would agree with us. In the revised manuscript, we properly cited Cipriani et al., (2021) (line 96-97, 287, 468-469 and 482).

2) All of the TURBO-ID strains have reduced viability. Since there is some concern that P granule composition could be affected in the tagged strains, showing the localization of other known components of P granules (GLH-1, PGL-1) in these lines is critical.

This is a great suggestion.

Using genetic crosses, we generated deps-1::TurboID (I), and TurboID::glh-1(I) strains expressing PGL-1::TagRFP(IV) or GFP:CSR-1(IV), two well-characterized P granule proteins. Images and description of new data are updated (Figure 1—figure supplement 1A and line 128-134).

3) The wording of "cytoplasmic vs perinuclear" P granule quantification a bit inaccurate – this should be modified.

We thank reviewers for raising this issue. We agree with reviewers that the term rachis is more accurate. “cytoplasmic vs. perinuclear” is replaced with “rachis vs. edge” in the main text, Figure legend and Materials and methods (line 299, 302, 305-307, 673, 676, 679, 875, 879, 891, and 905).

4) The authors should further discuss how many of the proteins identified by TurboID were previously identified and known components of P granules, nuclear pores, or other neighboring germ granule compartments.

We compiled a list of known components of P granules, Z granules and Mutator foci based on previous studies (Supplementary file 2) (Manage et al., 2020; Updike and Strome, 2010; Wan et al., 2018). Since we performed proximity labeling using adult hermaphrodites, we excluded known granule components found in embryos or males (such as ALG-3 and ALG-4), or any ambiguous proteins that may localize to processing bodies (Gallo et al., 2008). The main text is updated (line 170-179). We also updated the Venn diagram and used asterisks to denote known P granule and Z granule proteins (Figure 2B).

Reviewer #3 (Recommendations for the authors):It would be important to know how many outcrosses were done of the strains.

The strains were outcrossed once to wild type animals. We updated this information in Materials and methods (line 623-624 and 630).

It looks like some proteins were tagged N' terminal and some C' terminal but this is not reflected in the methods (were the C-terminal fusions really FLAG::AID::GFP::TEV? Or were they TEV::GFP::AID::FLAG?)

Thanks for pointing this out. Tagged proteins are named by convention: N-terminal tag::Protein::C-terminal tag. EGGD-1 protein is C-terminally tagged with GFP::TEV::FLAG::AID and EGGD-2 protein is N-terminally tagged with mCherry (line 590 and Key Resources Table).

In the discussion about the different types of granules in the germ line and potential overlap, it would be helpful to mention how many known components of these other granules were found in their studies.

This is a great suggestion. Please see the responses above.

Scale bars should be provided on all images.

In the revised manuscript, scale bars are added to Figure 3—figure supplement 1B-D and Figure 4—figure supplement 1B-C.

The naming of the gene should be approved by the WormBase and consistent with the other manuscript (now published in eLife).

Please see our response above.

In Figure 2B legend: Strain names should be italicized. Check the entire manuscript for correct strain designation and nomenclature.

Thanks for the suggestion. Strains names are italicized.

Figure 2B: purple text is hard to distinguish from black.

We changed the color of purple to a lighter shade.

Figure 2C: It is not clear if these the only core and filament proteins? If not, list all of the proteins and highlight those detected in the pull-down.

The proteins listed in Figure 2C are the only core and filament proteins enriched in both TurboID pulldown experiments. There are other core and filament proteins that are not enriched. We included a list of all nucleoporins and marked the ones that are enriched in the Supplementary file 3.

Figure 3A: Cannot understand what Figure 3A is from the legend. Instead of "gene name" in the table, how about "RNAi target"?

Thank you for the suggestion. We have changed “Gene name” to “RNAi target” in Figure 3A.

Figure 3B: If plg-1::turboID is sterile, can you confirm that pgl-1::TagRFP strain is not compromised by showing brood size.

The pgl-1::TagRFP strain has been widely used and does not appear to display any defect in fertility (Wan et al., 2018, Wan et al., 2021). Indeed we found that the brood size of pgl-1::tagRFP animals is ~263 progeny per worm (Figure 3F), which is comparable to the brood size of N2 animals reported by other groups.

Germ line atrophy characterization, Brood size counting, and Germ line Mortality Assay were performed in the pgl-1::TagRFP background. We updated this information in Materials and methods (line 584-586) and figure legend (line 883, 886-887 and 902).

Germ line is the noun form; germline is the adjective form

We have revised our manuscript based on this suggestion.

Figure S3C: It is unclear to me why in the images of the S3C that they is gut fluorescence in the last (few cells) category but not the other categories. Is this real signal? Also, in the atrophy image it appears that germline tissue does have a bend. Seeing these with Hoecst co-stain would be helpful to evaluate the full morphology. Also it is appears that in small germ lines, the loss is in the early germ cells and that there are cellularizing oocytes earlier in the gonad (based on nuclear size).

The reviewer is right that the signal in the last panel is from gut autofluorescence. It becomes more pronounced because the contrast in the original image was auto adjusted. In the revised figure, we applied the same threshold to this image as the others when adjusting contrast.

In the “atrophy” group of animals, the germ line does not have a bend. We feel that a combination of brightfield and fluorescence images is sufficient to illustrate the germline morphology.

Figure 6 legend is showing the head not the pharynx.

We agree with the reviewer that it is accurate to use “head”. “pharynx” is changed to “head” in the text and Figure 6 legend (line 967 and 970).

Figure 6A. The inset is labelled "DAPI" but the DAPI separation is not presented to prove that is a nucleus and this is an issue as there are other cells in which the nuclei appear void of the mCherry protein.

A DAPI separation is added to Figure 6.

The reviewer is right that some nuclei are void of fluorescent proteins. Two reasons may account for it: (1). GLH-1 and EGGD-1 are expressed under a muscle-specific promoter and some cells in the head are not muscle cells; (2). GLH-1 and EGGD-1 are expressed from semi-stable extrachromosomal arrays and thus transgenic animals are mosaic (line 611 and 651).

Line 277: period missing at end of paragraph.

We revised the sentence.